# Micro- and nano-porosity of the active Alpine Fault zone, New Zealand

Martina Kirilova[1], Virginia Toy[1,2], Katrina Sauer[1], François Renard[3,4], Klaus Gessner[5,6], Richard Wirth[7], Xianghui Xiao[8,9], and Risa Matsumura[2]

[1]Institut für Geowissenschafte, Johannes Gutenberg Universität-Mainz, J. J. Becher Weg 21D-55128, Mainz, Germany

[2]Department of Geology, University of Otago, PO Box 56, Dunedin 9054, New Zealand

[3]Department of Geosciences, The Njord Centre, University of Oslo, Oslo 0316, Norway.

[4]Université Grenoble Alpes, Université Savoie Mont Blanc, CNRS, IRD, IFSTTAR, ISTerre, BP53, 38041 Grenoble, France.

[5]Geological Survey of Western Australia, 100 Plain Street, East Perth, WA 6004, Australia.

[6]School of Earth Sciences, The University of Western Australia, 35 Stirling Highway, Crawley, WA 6009

[7]Helmholtz-Zentrum Potsdam, GFZ, Sektion 4.3, Telegrafenberg, 14473 Potsdam, Germany

[8]Advanced Photon Source, Argonne National Laboratory, Lemont, IL 60439, USA

[9]National Synchrotron Light Source II, Brookhaven National Laboratory, Upton, NY 11973, USA

*Correspondence to:* Martina Kirilova (martina.kirilova@uni-mainz.de)

## Abstract

Porosity reduction in rocks from a fault core can cause elevated pore fluid pressures, and consequently influence the recurrence time of earthquakes. We investigated the porosity distribution in the New Zealand's Alpine Fault core in samples recovered during the first phase of the Deep Fault Drilling Project (DFDP-1B) by using two-dimensional nanoscale and three-dimensional microscale imaging. Synchrotron X-ray microtomography-derived analyses of open pore spaces show total microscale porosities in the range of 0.1-0.24%. These pores have mainly non-spherical, elongated, flat shapes and show subtle bipolar orientation. Scanning and transmission electron microscopy reveal the samples' microstructural organization, where nanoscale pores ornament grain boundaries of the gouge material, especially clay minerals. Our data imply that: (i) the porosity of the fault core is very small and not connected, (ii) the distribution of clay minerals controls the shape and orientation of the associated pores; (iii) porosity was reduced due to pressure solution processes; and (iv) mineral precipitation in fluid-filled pores can affect the mechanical behaviour of the Alpine Fault by decreasing the already critically low total porosity of the fault core, causing elevated pore fluid pressures, and/or introducing weak mineral phases, and thus lowering the overall fault frictional strength. We conclude that the current state of very low porosity in the Alpine Fault core is likely to play a key role in the initiation of the next fault rupture.

## 1. Introduction

Fault mechanics, fault structure and fluid flow properties of damaged fault rocks are intimately related (e.g. Gratier and Gueydan, 2007; Faulkner et al., 2010). Fault rupture is associated with intense brittle fracturing that enhances porosity, and thus permeability, and therefore also possible rates and directions of fluid propagation within fault zones (e.g. Girault et al., 2018). Conversely, post seismic recovery mechanisms (gouge compaction and pressure

solution processes) result in reductions of porosity, permeability and fluid flow (Renard et al, 2000; Faulkner et al.,
2010; Sutherland et al., 2012). These processes may cause elevated pore fluid pressures within fault cores, and
trigger frictional failure (e.g. Sibson, 1990; Gratier et al., 2003; Zhu et al., 2020). Therefore, the state of porosity
within rocks from fault cores can play a key role in fault slip.
The Alpine Fault of New Zealand is late in its seismic cycle (Cochran et al., 2017), so studying it allows us to
investigate pre-earthquake conditions that may influence earthquake nucleation and rupture processes. Recently,
drilling operations were undertaken in this fault zone to investigate the *in situ* conditions (Sutherland et al, 2012,
2017). Slug tests in the DFDP-1B borehole (Sutherland et al., 2012) and laboratory permeability measurements of
core samples (Carpenter et al., 2014) indicate permeability decreases by six orders of magnitude with increasing
proximity to the fault. Furthermore, Sutherland et al. (2012) documented a 0.53 MPa fluid pressure difference across
the principal slip zone (PSZ) of the fault, which suggests that the fault core has significantly lower permeability than
the surrounding cataclasite units. It is therefore interpreted to act as a fault seal that limits fluid circulation within its
hanging wall (Sutherland et al., 2012). Permeability variations like this are closely associated with the porosity
evolution of fault cores, and thus are likely to affect the fault strength and seismic properties (Sibson, 1990; Renard
et al., 2000; Gratier and Gueydan, 2007).
In this study, we investigate the porosity distribution in rocks from the Alpine Fault core and consider the potential
effects of this porosity on fault strength. We have measured open pore spaces in these rocks from X-ray computed
tomography (XCT) datasets and examined pore morphology by implementing quantitative shape analyses.
Lithological and microstructural characteristics of these samples were performed by using scanning electron
microscopy (SEM) and transmission electron microscopy (TEM).
**2.    Geological setting**
New Zealand`s Alpine Fault (Fig. 1a) is a major active crustal-scale structure that ruptures in a large earthquake
every $291 \pm 23$ years, the last one of which occurred in 1717 (Cochran et al., 2017). The fault is the main constituent
of the oblique transform boundary between the Australian Plate and the Pacific Plate, accommodating around 75%
of the relative plate motion. Ongoing dextral strike-slip at $27 \pm 5$ mm yr$^{-1}$ along the fault has resulted in a total
strike-separation of ~ 480 km over the last 25 Ma (Norris and Cooper, 1995, 2001; Norris and Toy, 2014). In
Neogene time, a dip-slip component added to the fault motion has resulted in more than 20 km of vertical uplift of
the hanging wall (Norris and Cooper, 1995, 2001; Norris and Toy, 2014). Consequently, rocks comprising the
hanging wall of the fault have been exposed in various outcrops, where they can be studied in detail. The
amphibolite facies Alpine Schist is the metamorphic protolith of a ~ 1 km thick mylonite zone, which has been
exhumed from depth and now structurally overlies an up to 50 m thick zone of brittlely deformed cataclasites and
gouges (e.g. Norris and Cooper, 1995, 2001; Norris and Toy, 2014). These rocks have been investigated in outcrops
and from samples collected in three boreholes during the two phases of the Deep Fault Drilling Project (DFDP-1A,
DFDP-1B and DFDP-2B; Fig. 1a) along the Alpine Fault (Sutherland et al., 2012; Toy et al., 2015; Toy et al., 2017).
Most of the brittle shear displacement along the fault has been accommodated within the fault core, which includes
Principal Slip Zone (PSZ) gouges and cataclasite-series rocks (Toy et al., 2015). Both in surface outcrops and drill
core samples, the Alpine Fault manifests as a thin (5 to 20 cm thick) gouge zone with a predominantly random fabric
of clay-rich material (Toy et al., 2015; Schuck et al., 2020). This cohesive but uncemented layer has a grain size
significantly finer than the surrounding cataclasite units, which shows that the material was reworked only within
this layer, most probably as a result of ultra-comminution due to multiple shear events under brittle conditions
(Boulton et al., 2012; Toy et al., 2015). The local presence of authigenic smectite clays (Schleicher et al., 2015) and
calcite and/or chlorite mineralization within sealed fractures and in the gouge matrix (Williams et al, 2017) indicate
that mineral reactions are restricted to an alteration zone within the fault core (Sutherland et al., 2012; Schuck et al.,
2020). The Alpine Fault core has been interpreted to have formed during a cyclical history of mineralization, shear,
and fragmentation (Toy et al., 2015). In addition, in the DFDP-1B borehole (Fig. 1b, Sutherland et al., 2012) fault
gouges occur at two distinct depths: 128.1 m (PSZ-1) and 143.85 m (PSZ-2), which shows that the slip was not
localized within a single gouge layer (Toy et al., 2015).

## 3.   Sample description and analytical methods

### 3.1 Samples

Porosity analyses were performed on four samples representing PSZ gouges and cataclasites of the Alpine Fault
core, which were recovered from the DFDP-1B borehole (Fig. 1b, c; Sutherland et al., 2012). These are DFDP-1B
58_1.9, DFDP-1B 69_2.48, DFDP-1B 69_2.54 and DFDP-1B 69_2.57. Sample nomenclature includes drill core run
number, section number, and centimeters measured from the top of each section. These samples were recovered
from drilled depth of 126.94 m, 143.82 m, 143.88 m and 143.91 m, respectively.
Detailed lithological and microstructural descriptions of the DFDP-1B drill core were carried out simultaneously
with, and after the drilling operations by the DFDP-1 Science Team, and these data were later summarized by Toy et
al. (2015). Samples DFDP-1B 58_1.9 and DFDP-1B 69_2.48 belong to foliated cataclasite units (Fig. 1b, c; Toy et
al., 2015), described as ultracataclasites with gouge-filled shear zones located above PSZ-1 and PSZ-2 respectively.
Sample DFDP-1B 69_2.54 represents the gouge layer that defines PSZ-2, whereas sample DFDP-1B 69_2.57 is
composed of brown ultracataclasites that belong to the lower cataclasite unit (Fig. 1b, c; Toy et al., 2015).

### 3.2 X-ray computed tomography (XCT)

We imaged the samples using X-ray absorption tomography, where the signal intensity depends on how electron
density and bulk density attenuate a monochromatic X-ray along its path through the material (e.g. Fusseis et al.
2014). We acquired the X-ray microtomography data for this study at the 2-BM beamline of the Advanced Photon
Source, Argonne National Laboratories USA in December 2012. The non-cylindrical samples of ~7 mm height and
~ 4 mm diameter were mostly drilled parallel to the foliation, mounted on a rotary stage, and imaged with a beam
energy of 22.5 keV. A charge-couple device camera collected images at 0.25° rotation steps over 180°. A sample-
detector distance of 70 mm yielded a field-of-view of 2.81 mm. The voxel size (i.e. spatial sampling) was 1.3 µm
and the spatial resolution ranged from two to three times the voxel size. We have reconstructed the datasets with a
filtered back-projection parallel beam reconstruction into 32-bit gray level volumes consisting of 2048 * 2048 *
2048 voxels using X-TRACT (Gureyev et al., 2011).

### 3.3 Analyses of XCT datasets

Data analyses and image processing were performed using the commercial software Avizo 9.1™ (Fig. 2). Initially,
the datasets were rescaled to 8-bit grey scale volumes for enhanced computer performance. In addition, small
volumes of interest were cropped from the whole volume before a non-local means filter was applied to reduce noise
(Buades et al., 2005). For each voxel, this filter compares the value of this voxel with all neighboring voxels in a
given search window. A similarity between the neighbors determines a correction applied to each voxel (e.g.
Thomson et al., 2018).
On the filtered gray-scale images, pores were identified as disconnected materials of the darkest grey-scale range
(Fig. 2a, Supplementary material 1: Fig. 1). The corresponding gray-scale values were thresholded, and the datasets
were converted into binary form. This step is called segmentation. Several segmentation techniques exist, from
thresholding at a given gray scale value (e.g. Ianossov et al., 2009; Andrew et al., 2013) to deep learning algorithms
(Ma et al., 2020). It is up to the user to choose the segmentation technique that is most appropriate to analyze a given
dataset. To our knowledge, no single segmentation technique can be generalized and universally used independently
of the nature of the samples. In the present study, we have chosen a simple segmentation technique by applying a
threshold to the gray scale images to separate the void space from the solid. This technique has been used in many
studies in the last two decades to characterize porosity in rocks, including some very recent studies in rock physics
(Macente et al., 2019; Renard et al., 2019). The segmented porosity volume depends strongly on the choice of the
threshold and some studies have demonstrated that the final porosity estimated by different segmentation methods
can vary by 20% (Andrä et al., 2013). However, when the level of noise in the data is low, the differences in
porosities estimated by different segmentation techniques is negligible (Andrew, 2018). Our data were acquired at a
synchrotron where the parallel beam and high photon flux ensured a low level of the noise in the images. In
addition, application of a non-local-means filter applied to our data reduced the noise level. For these reasons, we
consider that it was robust to apply a simple thresholding technique to this dataset but acknowledge that the porosity
values we estimate could differ by <20% from the 'true' porosity of the rock (cf. Andrä et al., 2013; Hapca et al.,
132 2013).

However, our segmentation procedure also captured cracks within a sample, which are likely to result from
depressurization during core recovery (Fig. 2b, Supplementary material 1: Fig. 1). To omit the cracks, we utilized
the morphological operation 'connected components' available in the software Avizo 9.1, which allows volumes
larger than selected number of connected voxels to be excluded from the binary label images. To each sample we
applied upper limits of 20 (43.94 $\mu m^3$), 50 (109.85 $\mu m^3$), 100 (219.7 $\mu m^3$) and 200 (439.4 $\mu m^3$) face connected
voxels. Total porosities estimates based on these operations are presented as percentages of the sample volume in
Supplementary material 1: Table 1. Unfortunately, this methodology results in either loss of larger pores or inclusion
of small cracks depending on the implemented limit of connected components, and thus the calculated porosities
include significant bias. Therefore, the operation 'connected components' was used only for visualization purposes,
and clusters of 200 face connected voxels were created to show the 3D volumes of segmented pore spaces (Fig. 2c)
Instead, the volumes and shape characteristics of segmented materials (including cracks i.e. without any data
limitation) were exported from Avizo software in numerical format, and volume distributions within a sample were
plotted on a logarithmic scale (Fig. 3). Data up to a specific volume size were fit to a polynomial curve, and then the
curve was extrapolated to the X-axis intercept, which is the expected maximum pore size (Fig. 3). For each sample
the total porosity was then estimated by integrating the curve, which excludes all volumes on the right side of the
curve. Total porosities are presented as a percentage of the whole sample volume (Fig. 3). The implemented
equations are given in Supplementary material 1.
Pore shapes were analyzed on bivariate histograms plotted by using the numerical pore characteristics, previously
extracted from Avizo software. Only pore volumes between 21.97 $\mu m^3$ (10 voxels) and 878.8 $\mu m^3$ (400 voxels) were
included to avoid bias in the data due to insufficient voxel count and presence of cracks, respectively. Individual
pores in our dataset are separated (Fig. 2c).The covariance matrix of each pore was calculated, and the three
eigenvalues of this covariance matrix were extracted. These three values correspond to the three main orthogonal
directions in each pore (i.e. the longest, medium and shortest axes) and we use them as proxies to describe pore
geometry. Thus, their amplitudes provide information on the spatial extension of a given pore and its shape. The
ratio between the medium and largest eigenvalues of each pore defines its elongation (Fig. 4), the ratio between the
smallest and the largest eigenvalues defines its sphericity (Fig. 5), and the ratio of the smallest and the medium
eigenvalues defines its flatness (Fig. 6).
The angles θ and φ that describe the orientation of the longest eigenvalue (i.e. axis) of each pore with respect to the
global orthogonal axes system of the 3D scan were calculated. These angles were translated into trend and plunge
and then plotted on a lower hemisphere equal area stereographic projection with a probability density contour to
display the distribution of pore unit orientations (Fig. 7).
**3.4 Scanning electron microscopy (SEM)**
SEM images were collected on Zeiss Sigma-FF-SEM at the University of Otago's Centre for Electron Microscopy.
The SEM was operated at a working distance of 8.5 mm, a**n accelerating** voltage of 10 keV and a 120 μm aperture
with dwell time of 100μs. EDS maps were created by using Aztec Software ([https://www.oxford-](https://www.oxford-)
[instruments.com/products/microanalysis/energy-dispersive-x-ray-systems-eds-edx/eds-for-sem/eds-software-aztec](https://www.oxford-instruments.com/products/microanalysis/energy-dispersive-x-ray-systems-eds-edx/eds-for-sem/eds-software-aztec)).
**3.5 Transmission electron microscopy (TEM)**
TEM images were collected on a FEI Tecnai G2 F20 X-Twin transmission electron microscope, located at the
German Research Centre for Geosciences (GFZ), Potsdam, Germany (Fig. 9). The instrument is equipped with field-
emission gun (FEG) electron source and high-angle annular dark-field (HAADF) Detector. Images were collected
from samples placed on a Gatan double-tilt holder at an accelerating voltage of 200kV. These TEM samples were
prepared by focused ion beam (FIB) milling at GFZ Potsdam using a HELIOS system operated at an accelerating
voltage of 30 kV.

## 4    Results

### 4.1  XCT-derived characteristics of porosity

All samples contain low total porosities, ranging from 0.1% to 0.24% (Fig. 3). If different segmentation techniques
were applied, a variability in the range that Andrew (2018) demonstrated is reasonable, from nearly 0% to 20%,
would correspond to porosities between 0.08% and 0.29% in our samples. It can be noted that the lower cataclasite
sample (DFDP-1B 69_2.57) has twice as much pore space (Fig. 3d) as any of the other samples. The characterized
pore volume distributions range over almost three orders of magnitude for all samples (Fig. 3). Furthermore, the
expected maximum pore volume was estimated to be largest in the PSZ-2 sample (DFDP-1B 69_2.54), reaching 862
$\mu m^3$ (Fig. 3c).
In all samples, shape analyses of pores with volumes between 21.97 $\mu m^3$ (10 voxels) and 878.8 $\mu m^3$ (400 voxels)
demonstrate predominantly elongated (Fig. 4), non-spherical (Fig. 5) and flat pore shapes (Fig. 6). This is
particularly pronounced for the smaller pore volumes. The number of elongated pores per sample increases in the
upper foliated cataclasites (Fig. 4a and b) with increasing proximity to PSZ-2, where most elongated pores occur
(Fig. 4c). Conversely, the lower cataclasite sample demonstrates proportionally fewer elongated pores within the
sample (Fig. 4d). The degree of sphericity is uniform for all samples, and pores appear as mainly non-spherical (Fig.
5). Few isolated spherical pores are manifested only by small pore volumes (Fig. 5). A trend of increasing the
number of flat pores is observed with increasing sample depth (Fig. 6), and most flat pores are detected in the lower
cataclasite (Fig. 6d).
The orientations of the individual pore units show two distinctive peaks with opposite vergence, defining bipolar
distributions of pore orientations (Fig. 7). The observed bipolarity is subtle in samples DFDP-1B 58_1.9 (Fig. 7a)
and DFDP-1B 69_2.48 (Fig. 7b), and more obvious in samples DFDP-1B 69_2.54 (Fig. 7c) and DFDP-1B 69_2.57
(Fig. 7d).

### 4.2  Microstructural characteristics of porosity

To demonstrate the microstructural arrangement of the cataclasites, we show representative SEM images from
sample DFDP-1B 69_248 (Fig. 8), previously described as a 'lower foliated cataclasite' by Toy et al., 2015. SEM
images presented here reveal rounded to sub-rounded crystalline clasts up to 100 µm in diameter (Fig. 8a, b), which
consist of ~50 % plagioclase, ~40 % K-feldspar, and ~10 % quartz and are elongated at angles of 0-30° to the
foliation. The surrounding matrix material is composed of finer grains (< 30 µm in diameter) of white micas,
chlorite, K-feldspar, calcite and Ti-oxide (Fig. 8c). Numerous quartz clasts contain microfractures, filled by calcite
and/or chlorite.
TEM characterization of the gouge material from PSZ-2 (sample DFDP-1B 69_2.54) reveals that the Alpine Fault
gouges are composed of angular quartz and/or feldspar fragments (~200 nm in size), wrapped by smaller
phyllosilicates (< 100 nm long). This random fabric is ornamented by nanoscale pores (< 50 nm), distributed along
all grain and phase boundaries, but especially abundant within/around clay minerals (Fig. 9a).
The gouge material also demonstrates phyllosilicate-rich areas, defined by an increase in the clay/clast ratio. In these
zones, fine (< 100 nm long) and coarser (few µm long) clay grains coexist and are aligned in wavy fabric that
surrounds sporadic protolith fragments (Fig. 9b). Pore spaces are again distributed along the boundaries of the
constituent mineral grains but some of them are larger (~0.5 µm) with thin ellipsoidal or elongated shapes (Fig. 9b,
c). These pores are commonly associated with inter-clay layer porosity. Large size pores are also observed along
quartz-feldspar phase boundaries. These latter pores are associated with multiple grains and occasionally disrupt the
boundaries, thus were labelled as fracture porosity (Fig. 9d).
**5 Discussion**
**5.1 Characteristics of porosity within the Alpine Fault core**
Porosity analyses of samples from, or in close proximity to the two PSZs encountered in the DFDP-1B drill core
reveal total pore volumes between 0.1% and 0.24% (Fig. 3). These values are significantly lower than the porosity
estimates from other active faults in the world, such as: 0.2 to 5.7% total porosity in the core of the Nojima Fault,
Japan (Surma et al., 2003) and 0 to 18% in the San Andreas Fault core (Blackburn et al., 2009). The Alpine Fault
core contains total pore space volumes comparable only with the lower porosities in these previous studies. It should
be noted that the smallest pore spaces captured in the XCT datasets are 1.3 µm in size due to acquisition constraints,
whereas nanoscale porosity was identified on the TEM images. Therefore, the estimated total porosities from XCT
data represent only minimum values of the open pore spaces in the Alpine Fault core.
TEM images presented here mainly focus on nano-scale materials (Fig. 9a, c, d) but were also used to describe the
distribution of micro-porosity in these rocks (Figure 9b). The pores visible on grain and phase boundaries in figure
9b have similar sizes to the pores segmented on XCT images (> 1.3 µm in diameter), thus we conclude that this is
the typical habit of both nano- and micro-pores within the Alpine Fault core (Fig. 9). In addition, both quantitative
micro-porosity shape analyses (Fig. 4, 5 and 6) and nano-pores identified on TEM images (Fig. 9) reveal that a
significant population of pores are predominantly non-spherical with elongated, flat shapes. We attribute this
observation to the tendency of these pores to ornament clay minerals where pores are distributed and elongated
along their (001) planes (Fig. 9b, c and d).
Foliation in the upper cataclasites is defined by clay-sized phyllosilicates, that become more abundant with
proximity to the PSZ (Toy et al., 2015), where a weak clay fabric is developed (Schleicher et al., 2015). This gradual
enrichment in clay minerals coincides with the subtle development of bipolar distributions of pore orientations with
increasing sample depth (Fig. 7). This observation and the fact that pores are mainly distributed along grain
boundaries of clays (Fig. 9) suggest that the distribution of clay minerals also controls pore orientations within the
Alpine Fault core. Previously, the phyllosilicate foliation in the Alpine Fault cataclasites has been used to define
shear direction (Toy et al., 2015). Thus, we speculate that pore orientations in these rocks are also systematically
related to the kinematic framework of the shear zone. If these pores represent remnants of fluid channels, their
spatial orientation is likely to reflect the fluid flow directions during deformation. To address this possibility more
data for systematic analyses of pore orientations are needed.
**5.2 Porosity reduction within the Alpine Fault core**
The comparatively lower porosity estimates of the Alpine Fault core than other active faults (e.g. the Nojima Fault,
Surma et al., 2003, and the San Andreas Fault, Blackburn et al., 2009) could be attributed to the fact that the Alpine
Fault is late in its c. 300 year seismic cycle and the last seismic event occurred in 1717 (Cochran et al., 2017). Thus,
we propose that the fault has almost completely sealed. Porosity of fault cores is believed to evolve during the
seismic cycle, since fault rupture can cause porosities to increase up to 10% (Marone et al., 1990), and subsequent
healing mechanisms (such as mechanical compaction of the fault gouge and/or elimination of pore spaces within the
fault core due to pressure solution processes) cause porosity to decrease over time (Sibson, 1990; Renard et al.,
2000; Faulkner et al., 2010). SEM data presented here show that fine-grained chlorite and muscovite grains formed
as a cement in the cataclastic matrix (Fig. 8c). Our TEM data reveal the abundance of newly precipitated authigenic
clays, wrapped around coarser clay minerals (Fig. 9b). Furthermore, delicate clay minerals form fringe structures
(Fig. 9a), and strain shadows (Fig. 9c) around larger quartz-feldspar grains. These microstructural observations
demonstrate that pressure solution processes operated within these rocks (Toy et al., 2015).
Evidence for pressure solution processes has been previously documented in all units, comprising the Alpine Fault
core (Toy et al., 2015). Abundant precipitation of alteration minerals (Sutherland et al., 2012), calcite filled
intragranular and cross-cutting veins (Williams et al., 2017), and the occurrence of newly formed smectite clays
(Schleicher et al., 2015) indicate extensive fluid-rock reactions. In addition, anastomosing networks of opaque
minerals (such as graphite; Kirilova et al., 2017), which define foliation in the upper cataclasites (Toy et al., 2015),
have been interpreted to be concentrated by pressure solution processes during aseismic creep (Toy et al., 2015;
Gratier et al., 2011). The petrological characteristics of the Alpine Fault core lithologies indicate that solution
transfer was likely the dominant mechanism for pore closure within these rocks.
Porosity estimates presented here are so low that presumably negligible variations in between samples can represent
significant gradients in porosity. For example, the increase of total porosity in sample DFDP-1B 69-2.57 with only
0.14%, manifests as twice as many open pore spaces in comparison to the rest of the analyzed samples (Fig. 3). In
addition, this is the only footwall sample analyzed here and as already mentioned in section 3.1 does not contain any
gouge material. Post-rupture porosity reduction is known to operate three to four times faster within fine-grained
fault gouges than in coarser-grained cataclasites (Walder and Nur, 1984; Sleep and Blanpied, 1992; Renard et al.,
2000) which may explain the porosity differences demonstrated above. Furthermore, previous studies documented
less carbonate and phyllosilicate filling of cracks in the Alpine Fault footwall cataclasites than in the hanging wall
cataclasites (Sutherland et al., 2012; Toy et al., 2015), suggesting more reactive fluids are present and isolated
within the hanging wall of the Alpine Fault. Thus, more intense dissolution-precipitation processes took place in the
fault`s hanging wall, which very likely resulted in more efficient porosity reduction, as demonstrated by our porosity
estimates (Fig. 3).
**5.3 Effects of porosity on the Alpine Fault strength**
Very low porosity estimates are presented here (Fig. 3). Very low permeabilities of $10^{-18}$ m$^2$ were also measured
experimentally in clay-rich cataclasites and gouges from the Alpine Fault zone (Carpenter et al., 2014). In addition,
the documented difference of total porosities between the hanging wall and footwall samples (Fig. 3) may be
interpreted to reflect different intensities of pressure solution processes, and thus compartmentalization of
percolating fluids. Our porosity data show a spatial trend similar to the permeability measurements of Carpenter et
al. (2014). This observation yields increased confidence in the interpretation of Carpenter et al. (2014) of a
permeability gradient with distance from the PSZ, which itself acts as a hydraulic seal (Sutherland, et al., 2012). The
existence of such a barrier to flow is characteristic for faults undergoing creep and locked faults (Rice, 1992;
Labaume et al., 1997; Wiersberg and Erzinger, 2008). However, much higher permeabilities in the surrounding
damaged rocks (Carpenter et al., 2014) allow fast propagation of fluids within them and can cause localization of
high fluid pressures on one side or the other of a hydraulic seal (Sibson, 1990). Such fluid pressures can enhance
gouge compaction and pressure solution processes, which will eventually introduce zones of weakness and thus may
trigger fault slip (Faulkner et al., 2010).
Previous studies and the observations presented here show that fluids were present in the Alpine Fault rocks. Fluid-
filled pores represent a favorable environment for mineral precipitation, which can affect the fault strength in two
ways: (i) Very small decrease of these critically low total porosities due to mineral precipitation would cause fluid
pressurization, which is a well-known fault weakening mechanism described by Byerlee (1990) and Sibson (1990);
however, this pressure increase could be slightly offset by inclusion of fluids into new hydrous minerals; (ii)
deposition of frictionally weak phases (such as clay minerals and graphite), especially if they decorate grain contacts
and/or form interlinked weak layers, would lower the overall frictional strength (Rutter et al., 1976; Niemeijer et al.,
299 2010).

Precipitated authigenic clay minerals were identified in our TEM data (Fig. 9) and also documented by previous
studies (Schleicher et al., 2015). As well as having low frictional strengths (Moore and Lockner, 2004), clay
minerals may also contribute to the formation of an impermeable seal if they form an aligned fabric, which can
enhance the likelihood of fluid-pressurization in the fault rocks (Rice, 1992; Faulkner et al., 2010). In addition,
graphite, which was previously documented in these rocks (Kirilova et. al., 2017), may effectively weaken the fault
due to mechanical smearing (Rutter et al., 2013) and/or localized precipitation within strained areas (Upton and
Craw, 2008). Such graphite precipitation within shear surfaces was previously documented by Kirilova et al. (2017).
In summary, the presence of trapped fluids in the low porosity rocks of the Alpine Fault core possibly controls the
mechanical behavior of the fault and could be responsible for future rupture initiation due to fluid pressurization
and/or precipitation of weak mineral phases. This hypothesis is further supported by an experimental study showing
that the DFDP-1 gouges are frictionally strong in the absence of elevated fluid pressure (Boulton et al., 2014).

## 6  Conclusions

Analyses of XCT-datasets and TEM images of borehole samples from the core of the Alpine Fault reveal micro- and
nanoscale pores, distributed along grain boundaries of the constituent mineral phases, especially clay minerals. The
tendency of these pores to ornament clays defines their predominantly non-spherical, elongated, flat shapes and the
bipolar distribution of pore orientations. The documented extremely low total porosities (in the range 0.1-0.24 %) in
these rocks suggest effective porosity reduction and fault healing. Microstructural observations presented here and
documented in previous studies indicate that pressure solution processes were the dominant healing mechanism, and
that fluids were present in these rocks. Therefore, fluid-filled pores may be places where elevated pore fluid
pressures develop, due to further mineral precipitation that decreases the already critically low total porosities.
Alternatively, these pores may also facilitate the deposition of weak mineral phases (such as clay minerals and
graphite) that may very effectively weaken the fault. We conclude that the current state of the fault core porosity is
possibly a controlling factor on the mechanical behaviour of the Alpine Fault and will likely play a key role in the
initiation of the next fault rupture.

**Data availability.**

Avizo screenshots, total porosity estimates, Matlab script and numerical data of pore volumes can be found in
Supplementary material 1.

**Authors contribution**

Kirilova reconstructed, processed, and analysed the XCT datasets presented here, interpreted the TEM data and
prepared the manuscript. Most of this work was performed during Kirilova's PhD under the academic guidance of
Toy. Toy and Gessner collected the XCT data with technical support by Xiao. Renard and Sauer contributed with
valuable discussion about XCT data analyses and edited the manuscript. Wirth enabled TEM data acquisition and
provided his expertise on TEM data interpretation. Matsumura collected and analyzed the presented SEM data. The
final version of this manuscript benefits from collective intellectual input.

**Competing interests**

The authors declare that they have no conflict of interest.

**Acknowledgments**

We gratefully acknowledge funding from the Advances Photon Source (GUP 31177). This research used resources
of the Advanced Photon Source, a U.S. Department of Energy (DOE) Office of Science User Facility operated for
the DOE Office of Science by Argonne National Laboratory under Contract No. DE-AC02-06CH11357. Avizo
workstation was built at the University of Otago with financial support provided by Nvidia Corporation, Royal
Society of New Zealand`s Rutherford fellowships (16-UOO-001), the Ministry of Business and Innovation`s
Endeavor Fund (C05X1605/GNS-MBIE00056), and a subcontract to the Tectonics and Structure of Zealandia
Program at GNS Science (GNS-DCF00020). Publishing bursary funding provided by the University of Otago is
greatly appreciated. We thank Sherry Mayo for helping with the reconstruction process of XCT data and Andrew
Squelch for providing use of the Avizo workstation, located at CSIRO, Perth, Australia during the initial data
analyses. Special thanks to Reed Debaets for assistance with the development of Matlab code. Klaus Gessner
publishes with permission of the Executive Director, Geological Survey of Western Australia.

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

**Figures**

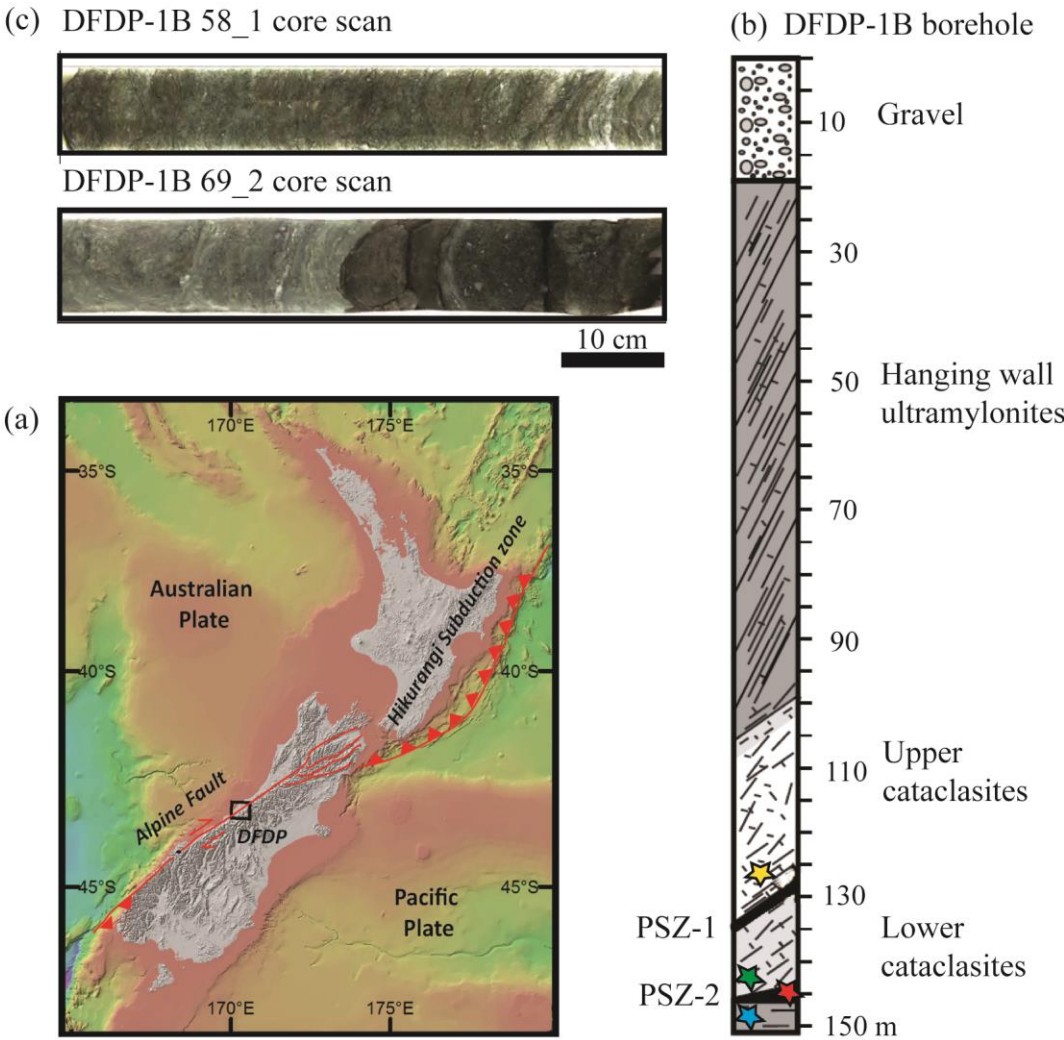


**Figure 1.** (a) Location map of DFDP drill sites (a bathymetric map compiled by NIWA). Drill site coordinates:
43°17′5″S, 170°24′22″E (b) Schematic diagram of the sampled lithologies in DFDP-1B borehole (modified after
Sutherland et al., 2012). (c) Scans of DFDP-1B drill core. Samples were collected from the locations indicated with
stars: yellow – DFDP-1B 58_1.9; green – DFDP-1B 69_2.48; red – DFDP-1B 69_2.54; blue – DFDP-1B 69_2.57.

DFDP - 1B 69-2.57

(a)

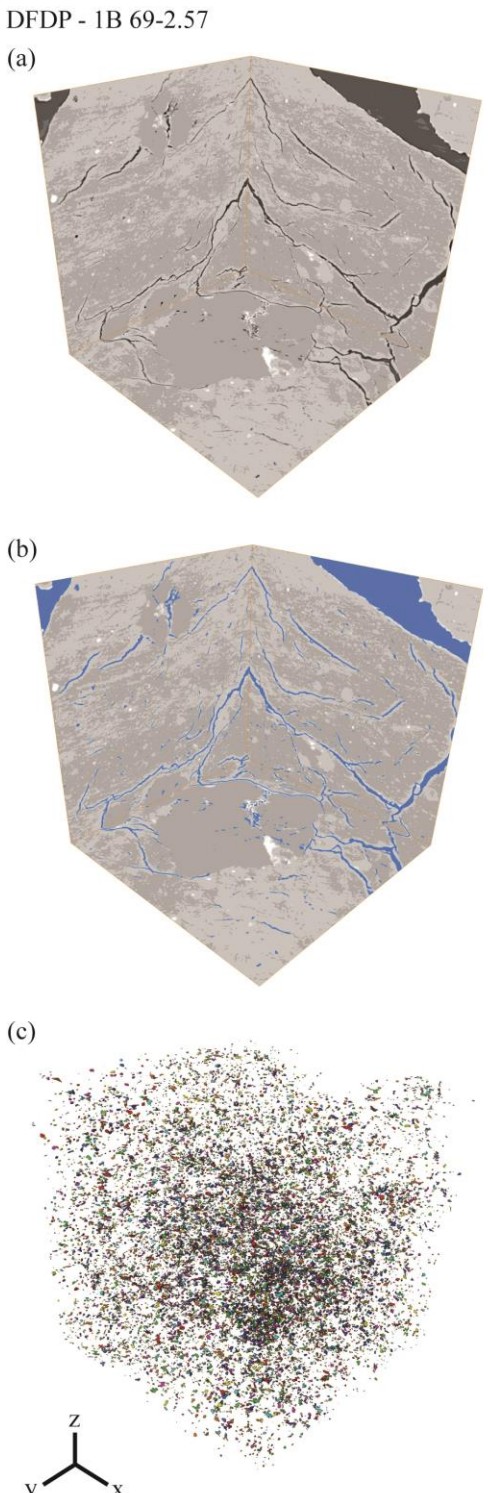

(b)

(c)


**Figure 2.** X-ray tomography data processing workflow. (a) Gray scale images in xy, xz and yz directions (b)
Threshold of the darkest gray scale phase in each sample, corresponding to voids (pores and fractures); (c) 3D
volume of the segmented pore spaces after removal of the fractures due to sample decompaction and coring
damaging effects.

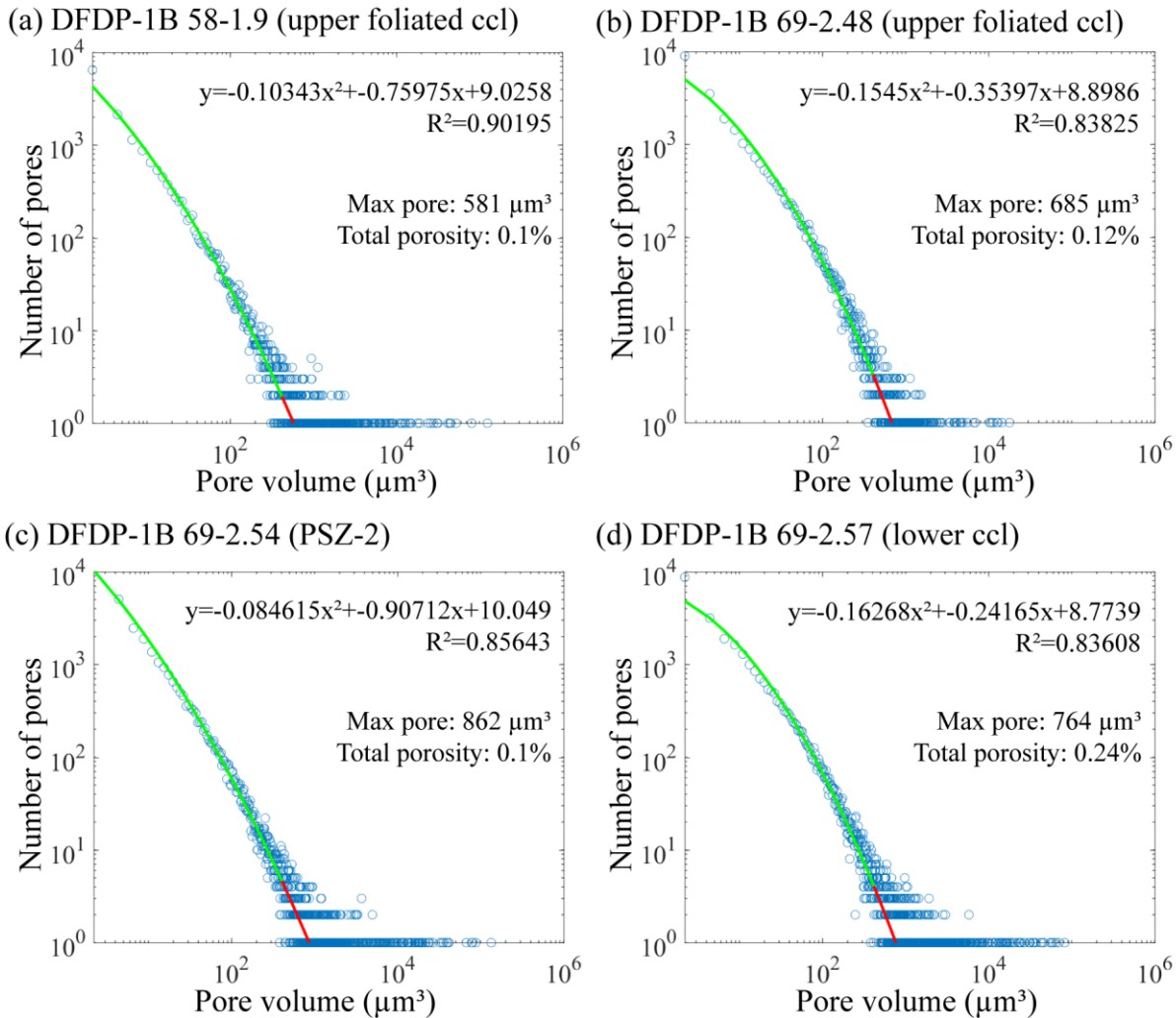

489

**Figure 3.** Plots of pore volume versus number of pores for each sample. Estimates of total porosity and size of the maximum expected pore are also shown, as well as the curve fitting function for each dataset.

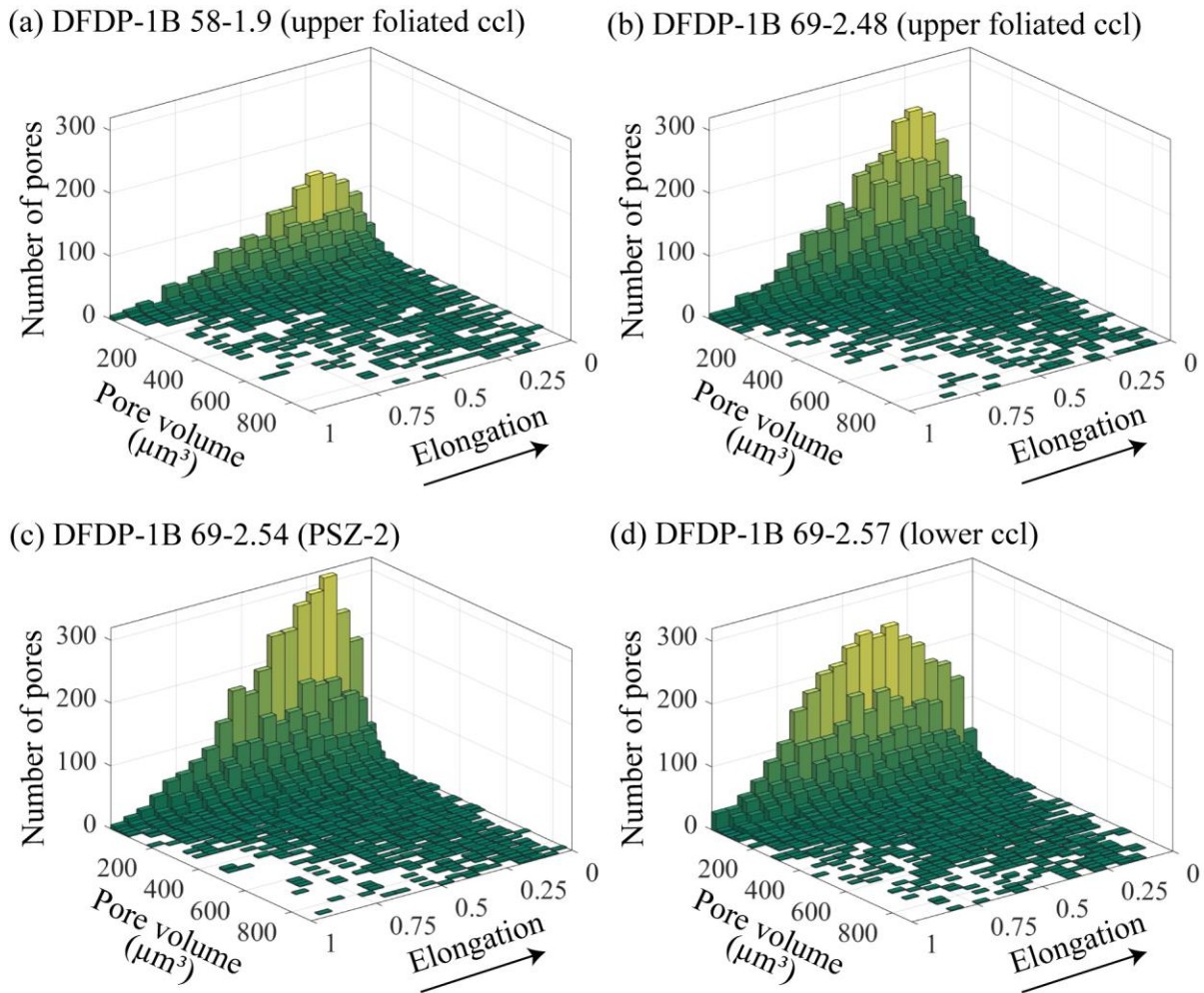

**Figure 4.** Bivariate histograms showing elongation versus pore volume ($\mu m^3$) and number of pores for each sample. The arrow indicates the direction of increasing elongation. Here, the elongation is defined as the ratio between the medium and the largest eigenvalues (i.e. axis) of each pore.

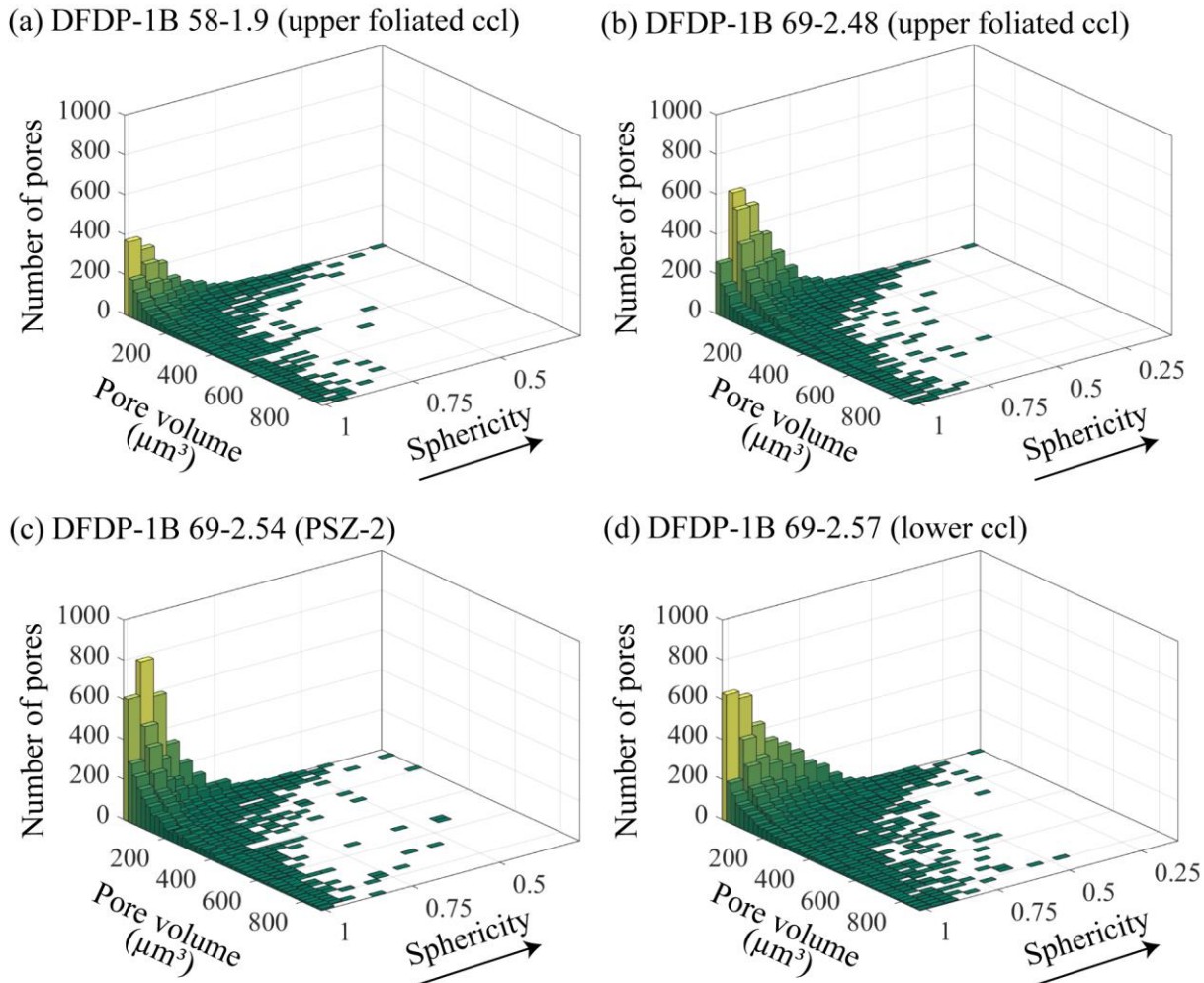


**Figure 5.** Bivariate histograms showing sphericity versus pore volume (μm³) and number of pores for each sample.
The arrow indicates the direction of increasing sphericity. Here, the sphericity is defined as the ratio between the
smallest and the largest eigenvalues (i.e. axis) of each pore.

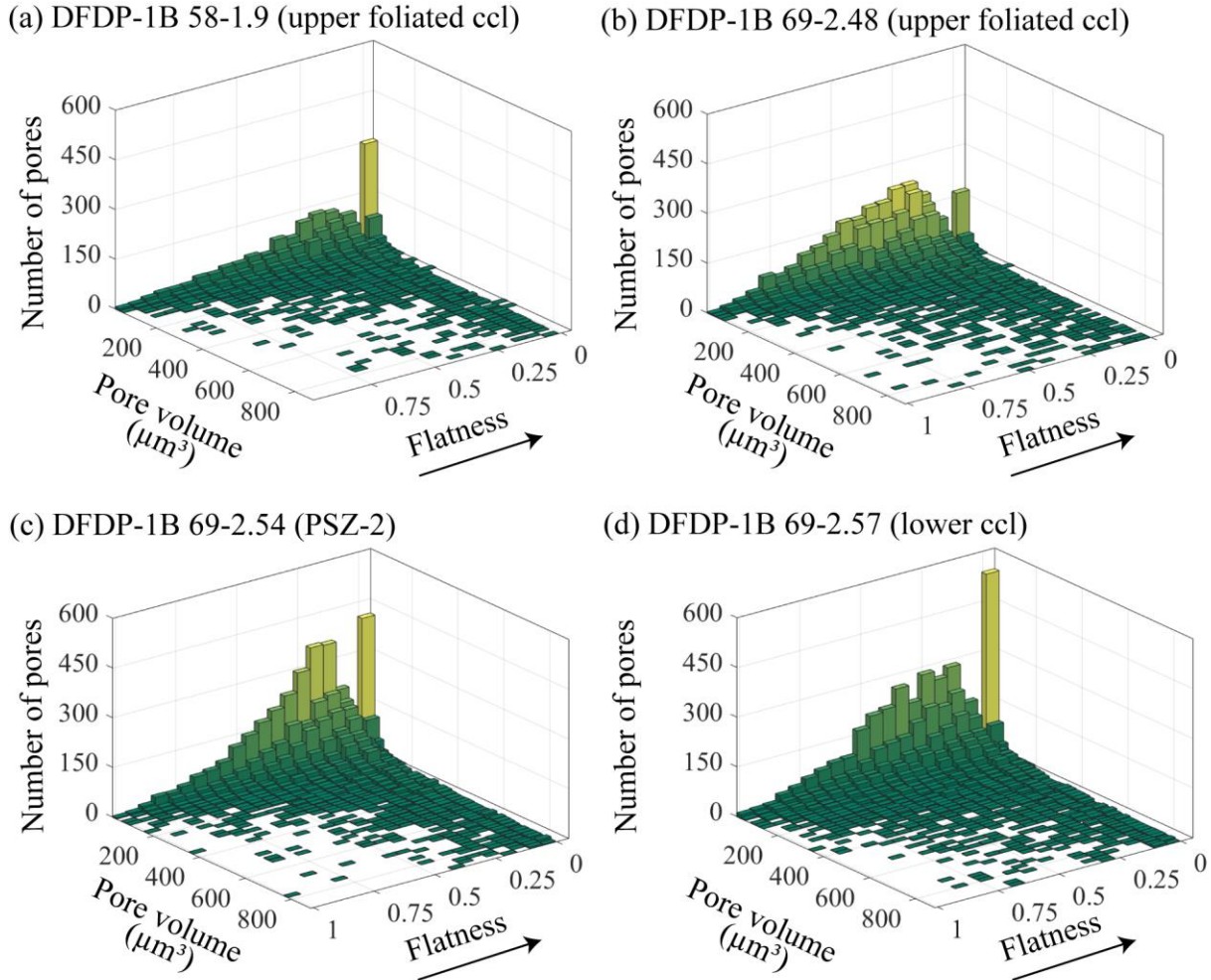


**Figure 6.** Bivariate histograms showing flatness versus pore volume ($\mu m^3$) and number of pores for each sample. The arrow indicates the direction of increasing flatness. Here, the flatness is defined as the ratio of the smallest and the medium eigenvalues (i.e. axis) of each pore.

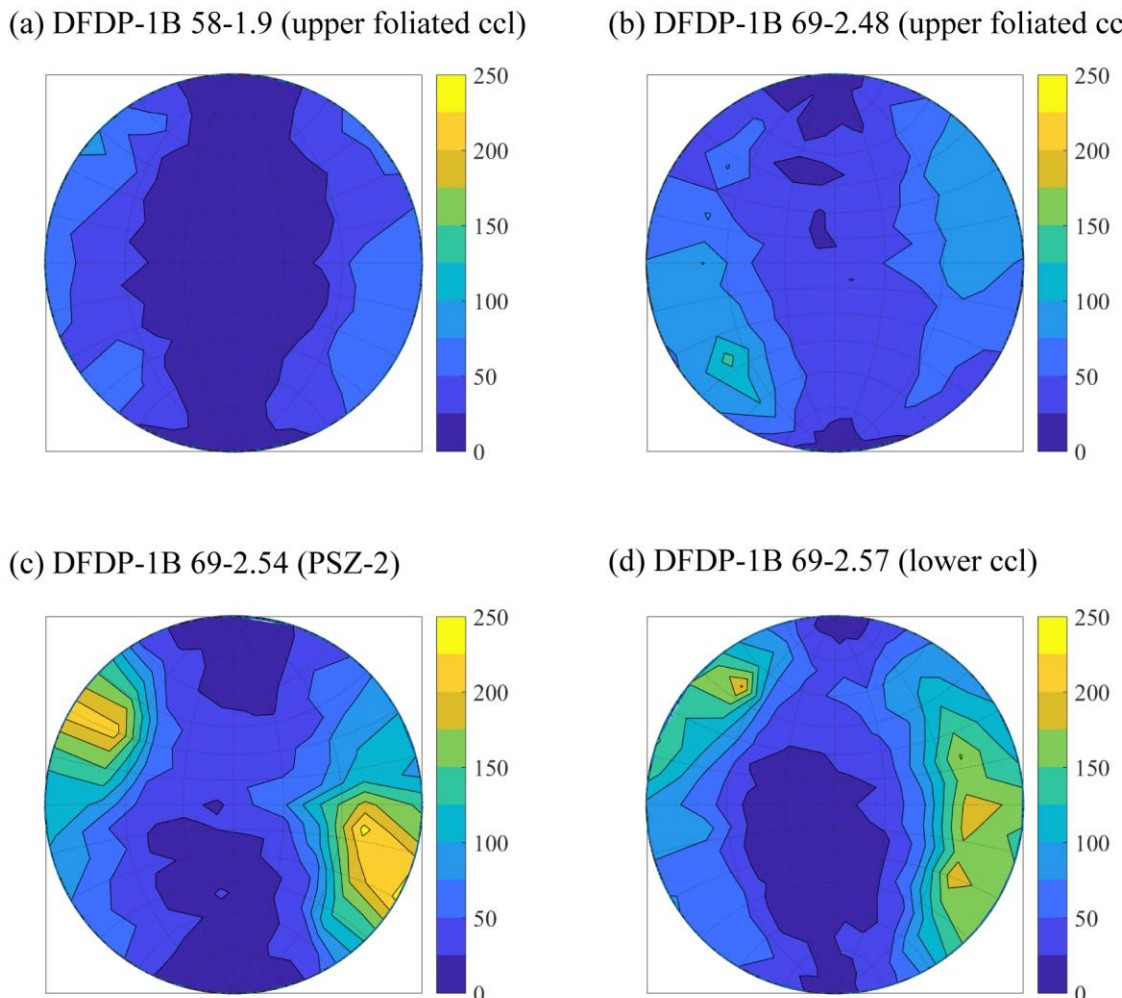


**Figure 7.** Distribution of pore unit orientations plotted on a lower hemisphere equal area stereographic projection with a probability density contour.


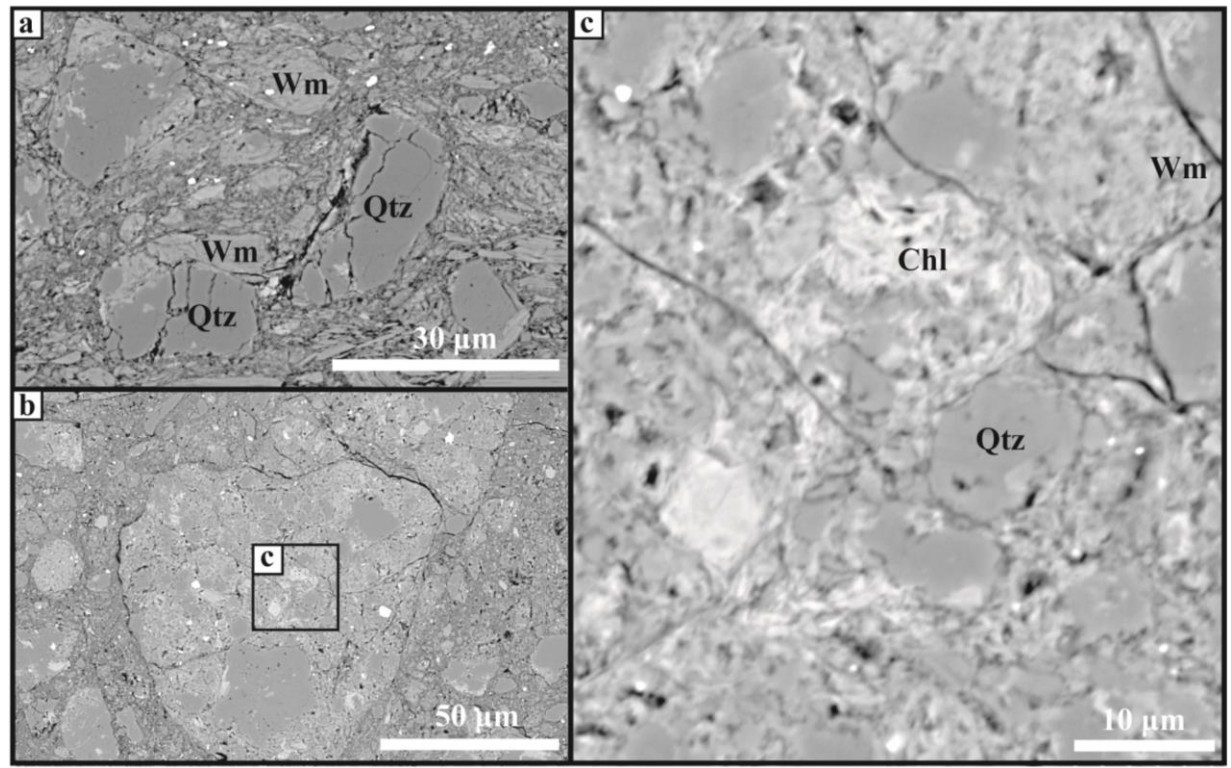

**Figure 8.** Scanning electron images collected from sample DFDP 1B 69-2.48 showing the existing mineral
associations. (a) Sub-rounded and intensly fractured quartz and white mica clasts, within fine matrix material. (b)
Reworked cataclasite clasts in phyllosilicate-rich matrix. (c) Fine chlorite and white mica aggregates between quartz
clasts. (Qtz = quartz, Wm = white mica, Chl = chlorite).

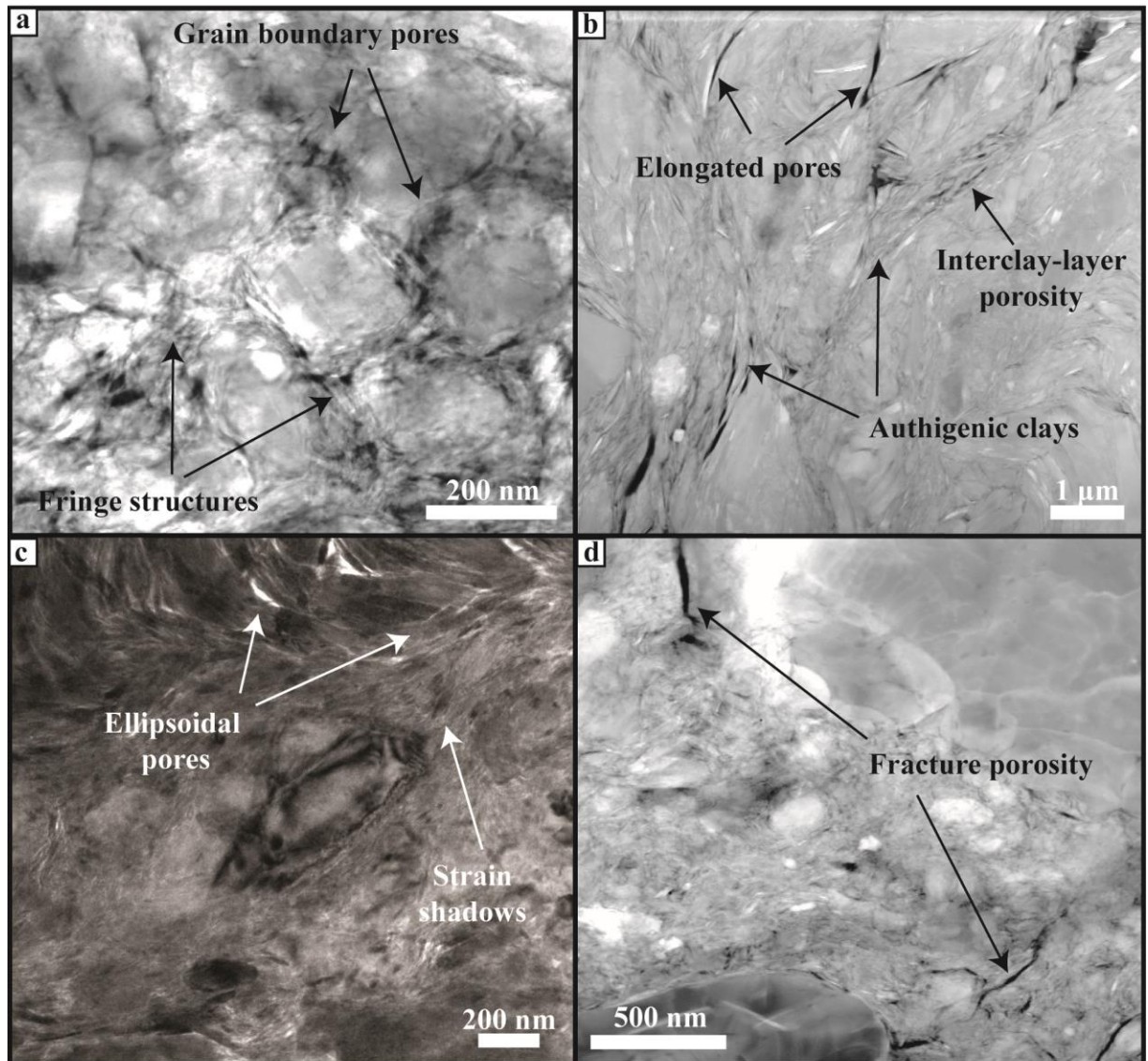

**Figure 9.** Transmission electron microscopy images collected from the gouge sample DFDP-1B 69_2.54 (PSZ-2).
(a) and (c) are bright-field (BF) images, where porosity appears as bright contrast areas. (b) and (d) are high-angle
annular dark field (HAADF) images, where pores appear as dark contrasts areas. (a) TEM bright-field image of
homogeneous fault gouge area. Quartz/feldspar grains, wrapped by fine authigenic clays, displaying fringe
morphlogies. Pores with sub-angular shape distributed along grain boundaries. (b) HAADF image of phyllosilicate-
rich gouge area. Co-existence of fine authigenic clays with coarser clay mineral grains. Elongated pores and
interlayer porosity. (c) TEM bright-field image of ellipsoidal pores in phyllosilicate-rich areas. Examples of strain
shadows along quartz/feldspar grains. (d) HAADF image of fracture porosity along grain boundaries of
quartz/feldspar grains.