# Peer review of "Micro- and nano-porosity of the active Alpine Fault zone, New 2 Zealand"

_Solid Earth, 2020_

## Referee Comment (RC1) · James Gilgannon (Referee) · 19 Jun 2020

Dear Editor,

As requested, I have reviewed the manuscript titled "Micro- and nano-porosity of the active Alpine Fault zone, New Zealand" by Kirilova et al., please find my general and specific comments below.

Kirilova et al. present data from the analysis of a core recovered during the Deep Fault Drilling Project (DFDP-1B). Synchrotron X-ray microtomography (XCT) was used in conjunction with Transmission electron microscopy (TEM) to gain information about the porosity and its habit during the critical pre-rupture stage of a major fault's seismic-cycle. These results are used to discuss the potential differences in processes that

are active in and around the Alpine fault and ultimately how this could effect a fault's transition to the next stage of its seismic cycle.

General comments:

The contribution from Kirilova et al. utilises a data set that is very special and provides a rare window into the microstructural state of a fault that is known to be approaching its next major earthquake rupture. Thus this contribution contains key information that brings us towards understanding how and why earthquakes can occur. The manuscript is well written, engaging and does a great job of orienting the reader in existing literature for hydro-chemo-mechanical feedbacks and the Alpine fault in general. It is clear that much effort has been made to make the contribution interesting and as such it was a great read.

This being said I have a few suggestions for the authors to consider in my specific comments. The majority of which relate to the methods section, where I think that some of the explanation should be reformulated and details added, alongside a suggestion of a figure to help the reader. Additionally, I think there is a need to better link the XCT and TEM data sets to enrich the results and hence shore up the discussion. I have one less trivial concern that I would like the authors to address: stated here briefly, I am uncertain about how much can be made of a difference of $\sim$0.14% porosity between samples given the filtering methods used. The authors use this difference as part of the main discussion on how changes in porosity will affect a major fault's mechanical state and I think that the significance of the result needs to be unpacked and evaluated more in the text. Lastly, I think the authors should be careful with when they say they 'demonstrate' or 'show' that certain processes are active. The results of this contribution are almost singularly observations about the characteristics of each sample's porosity and in this sense the authors do not show but rather interpret the presence of fluids or the activity of pressure solution. I think this is an important distinction to be drawn.

I very much enjoyed reading the manuscript and think that, with the inclusion of some more visualisation of the XCT data and some clarifications on the limitations of observations, it provides another solid step forward in understanding fault rocks and their dynamics.

Best wishes,

James Gilgannon

Specific comments:

I have tried to group the comments into blocks and they are ordered mostly in sequence with the order of the manuscript.

1. Comment on methods

As it stands I think the methods section needs bolstering in places. Below I have noted where I think the manuscript could benefit from this:

Lines 104 - 134: Analyses of XCT datasets

The structure here gives the feeling that you tired one method but subsequently chose another over it. After reading the manuscript over a few times I can see that this is not the case and you actually use both methods: in a first step, you use the 'connected components' method for visualising pores in space and then in a second step you characterise the porosity histograms with your MATLAB code.

I would recommend reformulating how section 3.3 is written to make it more clear that you did two things. I would go so far as to make subsections: 1) Segmented porosity for visualisation and 2) Quantifying total porosity. In this way it becomes clearer that you did both and the reason for using the integral of the pore volume histogram becomes clearer. Of course then you would require a further subsection for the description of pore geometry (ie. the use of the covariance matrix), of which I presumed you have used pores from the 'connected components' methods but limited to the size range you

stated.

Lines 123 - 130: Pore shape descriptors

The manuscript would significantly benefit from a figure illustrating the relevant aspects of the use of the covariance matrix. For example, I do not understand the author's characterisation of sphericity. I may have misunderstood the description but the ratio of two eigenvalues, which are both contained within a plane, surely cannot describe the deviation/tendency to a sphere, or have I misunderstood the metric you present? I am more familiar with sphericity being the ratio of the equivalent surface area of a sphere with the same volume as the pore volume over the actual surface area of the pore volume (e.g. Wadell, H. (1932))?

For this reason, I think that the section would benefit enormously from an example figure that corresponds to, and visualises the explanation of the metric. I imagine this would be best done with some specific examples of pore volumes from your data set. If the authors have not come up with the method themselves then I think that a citation for the more curious reader is also necessary.

Wadell, H., 1932. Volume, shape, and roundness of rock particles. The Journal of Geology, 40(5), pp.443-451.

Lines 132 - 134: Pore density calculation

I think that it should be briefly mentioned how the density calculation was made. Was a kernel used? and if so how was the bandwidth chosen to account for number of data points? Or was it a point density calculation, if so what neighbourhood was used? I think the amount of information currently given is too sparse.

2. Questions/concern regarding total porosity calculations

My questions/concern is regarding the uncertainty associated with the filtering of pore data used and how this translates into the discussed differences in the magnitudes of the total porosity from different samples. Your TEM results show that very small

fractures exist, which you identify as fracture porosity and, by the general argument of the paper, could have resulted from coring. While these fractures shown in fig. 8d are below the XCT resolution, I am brought to wonder how many slightly larger pores exist that are actually induced fractures. For example, the fact that so many small pores identified by XCT are almost completely flat in shape (fig. 6) might reflect that many small fractures, that are not syn-kinematic, are retained in the analysis. Therefore for me a question that presents itself is; does a simple size threshold, as you have used, have an appropriate amount of filtering information to allow a discussion about a difference of ∼0.14% porosity? Stated more plainly, how do you know if the variation between samples is not just a function of the degree to which each lithology experienced the coring and retrieval? Alternatively, can you rule out that the variation of ∼0.14% may just be related to the uncertainty of the polynomial fitting used to cap the pore size for integration?

I am uncertain if it is correct to straightway interpret this difference of ∼0.14% as meaningful. I think that more interrogation of this result needs to appear in the discussion before it is taken forward as independent confirmation of other literature. It might be that the authors wish to use the bore hole and laboratory measurements of permeability that are mentioned in the text to quantitatively check if the difference of ∼0.14% in total porosity can account for these differences in permeability. I am aware that this would require some assumptions when calculating but it would provide a base to the interpretation that a difference of ∼0.14% porosity between samples is meaningful. As currently presented I think that the result only convincingly shows that each calculated porosity is of the same order of magnitude.

3. Comment on linking XCT and TEM observations

The manuscript has a well crafted 'red thread' for the reader to follow but I feel that there is a gap in the current argument that requires some attention. The current formulation of the results goes from core/log scale to four very focused pictures of nano features by way of some abstract shape metrics at the micro scale. I am aware that figure 2

is supposed to bridge this gap by visualising the XCT data but it gives far too little information and doesn't allow the reader to see that your chosen TEM images are actually representative. The reader is left trusting the authors on things that can be evidenced with your current data sets.

To address this I think that there needs to be a more tangible link between the records of the microstructure in the XCT and TEM data sets. For example, the XCT and TEM data sets should be used for comparison/corroboration of the porosity/mineral associations. The XCT data is under-utilised with respect to showing the microstructure and the discussion would benefit from the evidential support that would come from the inclusion of a figure that visualises slices through the XCT data. In a very broad sense, this information showing what the microstructure looks like in the XCT data set is needed to provide a more convincing argument for the general habit of porosity (for example, that they occur 'especially' proximal to clay minerals). Currently, there are assumptions or logical jumps made by the authors in the discussion which are not necessary because the data sets at hand have information to support or falsify these suppositions. Additionally, the absence of this data was what partly led me to my comment/questions in point 2 because I was not given enough information to understand how the differences in total porosity estimates related to the different sample microstructures. Even with this aside, I would personally like to see a figure that better contextualises the micro-scale pores and their associations. Mostly I recommend this because, as I said in the general comments, your data sets are very special and as a curious reader I would like to be furnished with as much information of what the rocks look like as possible.

4. Question about section 5.3 and the concluding sentence of the manuscript

Is the porosity change not a consequence of the activity of other processes rather than a controlling factor? In the sense of your argument that the operation of mineral precipitation will lead to evaluated pore fluid pressures or fault rock weakness due to clay precipitation. Phrased as is, section 5.3 and the conclusions seem to make two arguments at the same time: the first giving the impression that porosity can provide

a driving force for change and the second that its change is just a marker for the increased activity of other processes which will drive change. I would argue, within the framework of your manuscript, that changes in porosity only chart the activity of other processes that actively dissipate energy and the activity of these other processes ultimately control fault rock stability.

5. Clarification of the word overpressure

As a last comment, I would recommend that the word overpressure is defined somewhere in the introduction. It is featured prominently in the first sentence and second last sentence of the abstract as well as the manuscript's conclusions but I am not sure to what the authors mean by it. I ask because it was my understanding that the bore hole fluid pressure measurements of Sutherland et al. (2012) found that, while fluid pressure was compartmentalised around the fault, the fluid pressure was never above hydrostatic. It may be worth a sentence or two that elaborates if the authors are referring to elevated fluid pressures or fluid pressures that exceed hydrostatic or some other meaning. Alternatively, the authors may not need to use the word overpressure as I think that the word is never mentioned in the discussion.

Sutherland, R., Toy, V.G., Townend, J., Cox, S.C., Eccles, J.D., Faulkner, D.R., Prior, D.J., Norris, R.J., Mariani, E., Boulton, C. and Carpenter, B.M., 2012. Drilling reveals fluid control on architecture and rupture of the Alpine fault, New Zealand. Geology, 40(12), pp.1143-1146.

Technical corrections:

Line 70: '. . .gouge zone with predominantly random fabric. . .' to '. . .gouge zone with a predominantly random fabric. . .'

Line 71: 'This cohesive but uncemented layer has significantly. . .' to 'This cohesive but uncemented layer has a significantly. . .'

Line 88: 'Detailed lithological and microstructural description. . .' to 'Detailed lithological

and microstructural descriptions...'

---

## Referee Comment (RC2) · Michel Bestmann (Referee) · 30 Jun 2020

In this research article the porosity distribution core samples of the Deep Fault Drilling project of the New Zealand Aloine Fault were investigated. Based on Synchrotron X-ray microtomography (3D data set) in combination with TEM analysis (2D data set) the porosity data were interpreted with respect of the permeability, fluid mobility and the possibility of fluid overpressure and their effects on the seismic cycle. The authors were able to show that fluid overpressurization in the Alpine Fault core controlled the mechanical behaviour of the fault and could be responsible for future rupture initiation.

The article is very well written and organized and provide very important data and interpretation to understand more in detail the processes, which control the seismic

cycle of active fault zones.

I only have minor comments on the manuscript

General comments:

1. Portion/fraction of weak minerals related to fluid overpressure in relation to pre-existent weak minerals (clay minerals in gouge zone / fault zone) - Abstract (line 25-29) and chapter 5.3 line 245-261:

You analyses gouge material, especialy clay minerals. In chapter 2. you mentioned that the gouge material is a reworked product probably as a result of ultracomminution due to multiple shear events under brittle conditions. The local presence of authigenic smectite clays (Schleicher et al., 2015) and calcite and/or chlorite mineralization within sealed fractures and in the gouge matrix (Williams et al, 2017) indicate that mineral reactions are restricted to an alteration zone within the fault core.

You conclude that due to fluid overpressure a weak mineral phase was introduced into the fault zone. My Question: What portion/fraction of the gouge material is related to the fluid overpressure and what part related to former events, e.g. ultracomminution together with fluid mobility/sealing, without fluid overpressure. Because when you already deal with a weak clay-rich rock and afterwards another weak phase in fluid-overpressurized pores is precipitated (e.g. clay, graphite), than the influence of this minor third weak phase (volume weighted with respect to the existing surrounding weak fault rock) on the already existent rheology is relatively low. Please clarify this point.

2. Analytical detection of 1 $\mu$m-sized pores

line 185-187: On figure 8b pores have sizes comparable to the small range of pores segmented on XCT images (> 1.3 $\mu$m in diameter), and thus we conclude that both nano- and micro pores within the Alpine Fault core are distributed on grain and phase boundaries, especially of clay minerals (Fig. 8).

MB-comment: In Fig. 8b the pores are extremely flattened /elongated and only the

long axis show a value > 1 $\mu$m. I am not sure if you can measure with synchrotron and a voxel size resolution of 1.3 $\mu$m such elongates pores where the calculated diameter is <1.3 $\mu$m. Please clarify this point.

Following points are minor comments:

Line 136: High resolution TEM images

MB-comment: Actually, your TEM images are not High Resolution images. The definition of high resolution TEM imaging means that you work with an Angstrom resolution in order make visible the atomic structure and that is not the case for your microstructures

line 200-201: To address this possibility more data for systematic analyses of pore orientations are needed

MB-comment: Please compare your observations/data with published papers, which contain similar TEM porosity analysis in clay-rich rocks.

line 229-231: Thus, the comparatively lower porosity estimates of the Alpine Fault core than other active faults (e.g. the Nojima Fault, Surma et al., 2003, and the San Andreas Fault, Blackburn et al., 2009) can be attributed to the fact that the Alpine Fault is late in its seismic cycle (Cochran et al., 2017).

MB-comment: Do you refer to the latest seismic event in the year 1717 and the average seismic cycle of 291 $\pm$ 23 years? please clarify this - because maybe the reader already forgot that you have mentioned this point at the begin of the paper

Figure 1 Line 403 (Figure 1):

MB-comment: Please provide GPS data of the drilling site

Figure 8 (a)

MB-comment: I presume the dark structures are the pores - please point directly with

[Figure]

arrow-tip onto the structures. Otherwise it is a little bit confusing, especially for readers who are not familiar in reading TEM images

Figure 8 (c)

MB-comment: where are the quartz/feldspar grains with the strain shadow? Is it the grain in the middle of the image? Than please shift the text "Ellipsoidal pores" to an area in the image where it does not cover an essential part of the image.

---

## Author Response (AR1)

Dear Dr. Florian Fusseis,

We are delighted to submit the revised version of our manuscript, entitled 'Micro- and nano-porosity of the active Alpine Fault zone, New Zealand' for consideration for publication in Solid Earth.

Herein, we investigated and analyzed the state of porosity within rocks from the fault core of a major active fault – the Alpine Fault of New Zealand. By the means of synchrotron X-ray microtomography and transmission electron microscopy, we acquired unique data obtained from samples recovered during the first phase of the Deep Fault Drilling Project (DFDP-1B). Our results demonstrate extremely low total porosities in these rocks, which we suggest were reduced due to post seismic recovery mechanisms. We conclude that these low porosities in the presence of fluids possibly control the mechanical behavior of the fault and could trigger a seismic slip along the fault due to fluid pressurization and/or precipitation of weak mineral phases.

We have revised the manuscript taking into account reviewers' comments. We have copied the reviews below and addressed each comment in turn. Our responses are indicated in green text. We used track changes to document the revisions to the manuscript and are resubmitting both a track changes, and changes accepted versions. We refer to revisions in the manuscript by line numbers – these are correct with respect to the revised version with changes accepted.

The affiliations of the corresponding author (Martina Kirilova) and Klaus Gessner have changed during the review process, thus those are now updated in the revised manuscript.

Thank you for your consideration of this manuscript.

Regards,
Martina Kirilova
Corresponding Author
martina.kirilova@uni-mainz.de
On behalf of the authors: Virginia Toy, Katrina Sauer, François Renard, Klaus Gessner, Richard Wirth, and Xianghui Xiao

As requested, I have reviewed the manuscript titled "Micro- and nano-porosity of the active Alpine Fault zone, New Zealand" by Kirilova et al., please find my general and specific comments below. Kirilova et al. present data from the analysis of a core recovered during the Deep Fault Drilling Project (DFDP-1B). Synchrotron X-ray microtomography (XCT) was used in conjunction with Transmission electron microscopy (TEM) to gain information about the porosity and its habit during the critical pre-rupture stage of a major fault's seismiccycle. These results are used to discuss the potential differences in processes that are active in and around the Alpine fault and ultimately how this could effect a fault's transition to the next stage of its seismic cycle.

General comments:

This being said I have a few suggestions for the authors to consider in my specific comments. The majority of which relate to the methods section, where I think that some of the explanation should be reformulated and details added, alongside a suggestion of a figure to help the reader. Additionally, I think there is a need to better link the XCT and TEM data sets to enrich the results and hence shore up the discussion. I have one less trivial concern that I would like the authors to address: stated here briefly, I am uncertain about how much can be made of a difference of 0.14% porosity between samples given the filtering methods used. The authors use this difference as part of the main discussion on how changes in porosity will affect a major fault's mechanical state and I think that the significance of the result needs to be unpacked and evaluated more in the text. Lastly, I think the authors should be careful with when they say they 'demonstrate' or 'show' that certain processes are active. The results of this contribution are almost singularly observations about the characteristics of each sample's porosity and in this sense the authors do not show but rather interpret the presence of fluids or the activity of pressure solution. I think this is an important distinction to be drawn.

Response: Thank you for this summary and the constructive comments. We have addressed each one of them following the specific comments below.

Specific comments:

I have tried to group the comments into blocks and they are ordered mostly in sequence with the order of the manuscript.

**1. Comment on methods**

As it stands I think the methods section needs bolstering in places. Below I have noted where I think the manuscript could benefit from this:

Lines 104 - 134: Analyses of XCT datasets

The structure here gives the feeling that you tired one method but subsequently chose another over it. After reading the manuscript over a few times I can see that this is not the case and you actually use both methods: in a first step, you use the 'connected components' method for visualising pores in space and then in a second step you characterise the porosity histograms with your MATLAB code.

I would recommend reformulating how section 3.3 is written to make it more clear that you did two things. I would go so far as to make subsections: 1) Segmented porosity for visualisation and 2) Quantifying total porosity. In this way it becomes clearer that you did both and the reason for using the integral of the pore volume histogram becomes clearer.

Response: We used Avizo software for initial processing (i.e. rescaling and filtering) of the data and thresholding. "Connected components" with limitation of the data (i.e. limiting the connected components up to 200 connected voxels) was applied only for visualization purposes. This is described in lines 105-114.

The remainder of the pore analyses (both total porosities and shape analyses) were performed by implementing Matlab scripts using the whole threshold range in numerical format (look at lines 116-117 (now in line 118) "Instead, the volumes of segmented materials (including cracks) were exported from Avizo software in numerical format"

We acknowledge the fact that the reviewer was confused about our methodology. Thus, in order to make the text easier to follow, in line 117 we added the following clarification:

"Therefore, "connected components" with limit of 200 connected voxels were used only for visualization purposes."

Of course then you would require a further subsection for the description of pore geometry (ie. the use of the covariance matrix), of which I presumed you have used pores from the 'connected components' methods but limited to the size range you stated.

Response: No, as stated in lines 116-117 ("Instead, the volumes of segmented materials (including cracks) were exported from Avizo software in numerical format") the pore sizes were not limited by the function "connected components". Any further data manipulation was performed on Matlab.

To clarify this, we changed the statement in lines 116-117 (now in line 118) to:

"Instead, the volumes and shape characteristics of segmented materials (including cracks) were exported from Avizo software in numerical format"

And also, in line 125 we replace "Pore shapes were analyzed on bivariate histograms." with "Pore shapes were analyzed on bivariate histograms plotted on Matlab by using the numerical pore characteristics, previously extracted from Avizo software."

Lines 123 - 130: Pore shape descriptors

The manuscript would significantly benefit from a figure illustrating the relevant aspects of the use of the covariance matrix. For example, I do not understand the author's characterisation of sphericity. I may have misunderstood the description but the ratio of two eigenvalues, which are both contained within a plane, surely cannot describe the deviation/tendency to a sphere, or have I misunderstood the metric you present? I am more familiar with sphericity being the ratio of the equivalent surface area of a sphere with the same volume as the pore volume over the actual surface area of the pore volume (e.g. Wadell, H. (1932))?

For this reason, I think that the section would benefit enormously from an example figure that corresponds to, and visualises the explanation of the metric. I imagine this would be best done with some specific examples of pore volumes from your data set. If the authors have not come up with the method themselves then I think that a citation for the more curious reader is also necessary.

Wadell, H., 1932. Volume, shape, and roundness of rock particles. The Journal of Geology, 40(5), pp.443-451.

Response: Thank you for this comment. No, we have not come up with the methods ourselves. All the shape analyses we performed are based on functions embedded in Avizo software that yield volumetric and shape characteristics for each segmented material in numerical format (lines 118). We simply plotted the results on bivariate histograms by using Matlab as stated in line 125.

We do not find it necessary to include in the manuscript a description of how the software produces those results as Avizo software is trusted source and every user/reader can refer to their library. However, here we provide a brief explanation of the functions we have used:

The covariance matrix is built on the basis of the moments of inertia and can be written as:

$$M = \left[ \begin{array}{ccc} M_{2x} & M_{2xy} & M_{2xz} \\ M_{2xy} & M_{2y} & M_{2yz} \\ M_{2xz} & M_{2yz} & M_{2z} \end{array} \right]$$

(matrix provided and implemented by Avizo software)

By using this matrix, the software computes the three eigenvalues by using a Singular Value decomposition. In an elongated ellipsoid the largest eigenvalue will describe the longest axis of the 3d object.

In this context, the deviation of the spherical form (i.e. anisotropy - a value extracted from Avizo software) is measured as 1 minus the ration of the smallest to largest eigenvalue. In a 3D object if the smallest and the longest axis are equal, the medium will have the same value as well, describing a spherical object and having numerical value = 0.

Lines 132 - 134: Pore density calculation

I think that it should be briefly mentioned how the density calculation was made. Was a kernel used? and if so how was the bandwidth chosen to account for number of data points? Or was it a point density calculation, if so what neighbourhood was used? I think the amount of information currently given is too sparse.

Response: We plotted the orientation of the longest eigenvalue of each pore on a lower hemisphere equal area stereographic projection. Thus, these stereonets do not represent pore density calculation but clusters of pores with preferred orientation. The data was plotted by using bivariate histogram bin counts implemented in Matlab (i.e. histcounts2), where:

[N,Xedges,Yedges] = histcounts2(X,Y,Xedges,Yedges)

The bivariate histogram results in bins with a predefined set of edges and the number of pore orientations that fall within each bin. This partitions X and Y into bins with the bin edges specified.

For contouring we used the countouring algorithm implemented in Matlab. Relevant lines from the script used are below:

% get histogram of resulting lats/longs and contour

[cs,cLats,cLons] = histcounts2(Lats,Lons,-95:10:95,-100:20:100); cLats(end)=[]; cLons(end)=[]; cLats = cLats + (cLats(2)-cLats(1))/2; cLons = cLons + (cLons(2)-cLons(1))/2;

% and plot it

COLORBAR = 0:25:225; contourfm(cLats,cLons,cs,'LevelList',COLORBAR(2):COLORBAR(2):COLORBAR(end)); caxis([COLORBAR(1) COLORBAR(end)])

contourcbar

2. Questions/concern regarding total porosity calculations

My questions/concern is regarding the uncertainty associated with the filtering of pore data used and how this translates into the discussed differences in the magnitudes of the total porosity from different samples. Your TEM results show that very small tures exist, which you identify as fracture porosity and, by the general argument of the paper, could have resulted from coring. While these fractures shown in fig. 8d are below the XCT resolution, I am brought to wonder how many slightly larger pores exist that are actually induced fractures. For example, the fact that so many small pores identified by XCT are almost completely flat in shape (fig. 6) might reflect that many small fractures, that are not syn-kinematic, are retained in the analysis. Therefore for me a question that presents itself is; does a simple size threshold, as you have used, have an appropriate amount of filtering information to allow a discussion about a difference of 0.14% porosity? Stated more plainly, how do you know if the variation between samples is not just a function of the degree to which each lithology experienced the coring and retrieval? Alternatively, can you rule out that the variation of 0.14% may just be related to the uncertainty of the polynomial fitting used to cap the pore size for integration?

I am uncertain if it is correct to straightway interpret this difference of 0.14% as meaningful. I think that more interrogation of this result needs to appear in the discussion before it is taken forward as independent confirmation of other literature. It might be that the authors wish to use the bore hole and laboratory measurements of permeability that are mentioned in the text to quantitatively check if the difference of 0.14% in total porosity can account for these differences in permeability.

~

I am aware that this would require some assumptions when calculating but it would provide a base to the interpretation that a difference of 0.14% porosity between samples is meaningful. As currently presented I think that the result only convincingly shows that each calculated porosity is of the same order of magnitude.

Response: We understand the reviewer's concern that big pores and small fractures could get easily misinterpreted/mislabeled in XCT datasets. This exactly is the prime reason why we decided against calculating total porosities in these samples by simply using 'connected components' and instead we fitted the data to a polynomial curve (mentioned in lines 115-123). We believe that implementing a mathematical approach is much more trustworthy than limiting the data based on the interpreter's bias. Furthermore, our total porosity calculations (by using the polynomial fit) roughly coincide with the total porosities yield by calculating the total porosities based on connected components with up to 200 voxels. You can see these numbers on the table below:

| DFDP-1B | polynomyal fit | 200 limit% |
|---|---|---|
| DFDP-1B 58_1.9 (Sam73) | 0.10 | 0.10 |
| DFDP-1B 69_2.48 (Sam79) | 0.12 | 0.11 |
| DFDP-1B 69_2.54 (Sam19) | 0.10 | 0.09 |
| DFDP-1B 69_2.57 (Sam69) | 0.24 | 0.17 |

The reviewer also expressed concern about the fact that some of the very flat pores may represent fractures. We acknowledge the validity of this statement. However, we believe that our approach of excluding cracks is efficient and possibly the best methodology for analyzing these samples (i.e. fitting the data to a polynomial curve). Furthermore, the shape of these pores is also very likely to result from their distribution along grain boundaries, especially of clay minerals (lines 190). The authors of the manuscript are in favour of this second scenario.

And last but not least, the difference of 0.14% of total porosity in these samples may seem insignificant to the reader. However, all of the samples contain extremely low porosities, and thus only 0.14% more pores actually result in doubling the amount of pores in sample DFDP-1B 69_2.57 in comparison to the rest of the samples. Thus, a discussion here is not only meaningful but also required, and very well related to changes in lithology in between the samples (lines 219-226, now in lines 222-229), and previous permeability measurements of these rocks (lines 233-236, now in lines 236-240).

3. Comment on linking XCT and TEM observations

The manuscript has a well crafted 'red thread' for the reader to follow but I feel that there is a gap in the current argument that requires some attention. The current formulation of the results goes from core/log scale to four very focused pictures of nano features by way of some abstract shape metrics at the micro scale. I am aware that figure 2 is supposed to bridge this gap by visualising the XCT data but it gives far too little information and doesn't allow the reader to see that your chosen TEM images are actually representative. The reader is left trusting the authors on things that can be evidenced with your current data sets.

To address this I think that there needs to be a more tangible link between the records of the microstructure in the XCT and TEM data sets. For example, the XCT and TEM data sets should be used for comparison/corroboration of the porosity/mineral associations. The XCT data is underutilised with respect to showing the microstructure and the discussion would benefit from the evidential support that would come from the inclusion of a figure that visualises slices through the XCT data. In a very broad sense, this information showing what the microstructure looks like in the XCT data set is needed to provide a more convincing argument for the general habit of porosity (for example, that they occur 'especially' proximal to clay minerals). Currently, there are assumptions or logical jumps made by the authors in the discussion which are not necessary because the data sets at hand have information to support or falsify these suppositions. Additionally, the absence of this data was what partly led me to my comment/questions in point 2 because I was not given enough information to understand how the differences in total porosity estimates related to the different sample microstructures. Even with this aside, I would personally like to see a figure that better contextualises the micro- scale pores and their associations. Mostly I recommend this because, as I said in the general comments, your data sets are very special and as a curious reader I would like to be furnished with as much information of what the rocks look like as possible.

Response: As we mentioned in lines 184-187 (now in lines187-190) the TEM images focus mainly on nano-scale materials, however, the largest pores observed on those images are also captured by (or comparable with) the smallest resolution of the XCT data. This justifies the validity of our argument that similar mineral – pore distribution is present both on nano- and macro-scale. This is further supported by the fact that both TEM (Fig. 8) and XCT shape analyses (Figs 4, 5 and 6) indicate the presence of predominantly elongated, flat pores (lines 188-191, now in lines191-194 ).

Therefore, we do not agree with the reviewer's comment that there is a gap in our arguments. Instead, we think we have provided sufficient data to demonstrate to the reader the validity of our interpretation rather than asking them to trust our judgement. Furthermore, we disagree that we have underutilized our XCT datasets. Instead, most of our interpretations are based on porosity estimates, and shape analyses yield from the XCT datasets. TEM images were merely used to relate the distribution of pores in respect to different minerals and to give a microstructural context to our porosity analyses.

Regardless, here we show two Avizo snapshots that demonstrate that pores are distributed along grain boundaries. The examples are taken from sample DFDP-1B 69_2.54.

[Figure]

Thresholded grey-grey scale image.

[Figure]

Grey scale image with segmented pores, marked by different colours.

4. Question about section 5.3 and the concluding sentence of the manuscript

Is the porosity change not a consequence of the activity of other processes rather than a controlling factor? In the sense of your argument that the operation of mineral precipitation will lead to evaluated pore fluid pressures or fault rock weakness due to clay precipitation. Phrased as is, section 5.3 and the conclusions seem to make two arguments at the same time: the first giving the impression that porosity can provide a driving force for change and the second that its change is just a marker for the in- creased activity of other processes which will drive change. I would argue, within the framework of your manuscript, that changes in porosity only chart the activity of other processes that actively dissipate energy and the activity of these other processes ulti- mately control fault rock stability.

Response: This is a very good point, thank you. In section 5.3, we aim to demonstrate to the reader that porosity is very closely interlinked with fluid circulation and mineral precipitation, both of which may change the mechanical behaviour of the rocks, and thus trigger an earthquake. However, the amount of porosity and/or the presence of porosity in these rocks defines how these processes may evolve. Therefore, the state of porosity in these rocks plays a key role, and thus we conclude that the porosity is actually a controlling factor on the mechanical behaviour of the Alpine Fault.

5. Clarification of the word overpressure

As a last comment, I would recommend that the word overpressure is defined some- where in the introduction. It is featured prominently in the first sentence and second last sentence of the abstract as well as the manuscript's conclusions but I am not sure to what the authors mean by it. I ask because it was my understanding that the bore hole fluid pressure measurements of Sutherland et al. (2012) found that, while fluid pressure was compartmentalised around the fault, the fluid pressure was never above hydrostatic. It may be worth a sentence or two that elaborates if the authors are re- ferring to elevated fluid pressures or fluid pressures that exceed hydrostatic or some other meaning. Alternatively, the authors may not need to use the word overpressure as I think that the word is never mentioned in the discussion.

Sutherland, R., Toy, V.G., Townend, J., Cox, S.C., Eccles, J.D., Faulkner, D.R., Prior, D.J., Norris, R.J., Mariani, E., Boulton, C. and Carpenter, B.M., 2012. Drilling reveals fluid control on architecture and rupture of the Alpine fault, New Zealand. Geology, 40(12), pp.1143-1146.

Response: We are familiar with the work of Sutherland et al. (2012), and we do agree with it. In addition, our work further supports the conclusions in their study (lines 237-239, now in lines 240-242). We do not state anywhere in our manuscript that fluid overpressure has been achieved in these rocks. We only speculate that the very low total porosities in these rocks and the processes affected by them (i.e. mineral precipitation and fluid circulation) can eventually lead to fluid overpressure, and thus trigger an earthquake. But in order to avoid confusion caused by different terminology, we will replace "fluid overpressure" with "elevated pore fluid pressures". (lines 17, 27, 272)

Technical corrections:

Line 70: '…gouge zone with predominantly random fabric…' to '…gouge zone with a predominantly random fabric…'

Response: Thank you. We will modify the text.

Line 71: 'This cohesive but uncemented layer has significantly…' to 'This cohesive but uncemented layer has a significantly…'

Response: The correction will be introduced in the text.

Line 88: 'Detailed lithological and microstructural description…' to 'Detailed lithological and microstructural descriptions. . .'

Response: We will modify the text accordingly.

In this research article the porosity distribution core samples of the Deep Fault Drilling project of the New Zealand Aloine Fault were investigated. Based on Synchrotron Xray microtomography (3D data set) in combination with TEM analysis (2D data set) the porosity data were interpreted with respect of the permeability, fluid mobility and the possibility of fluid overpressure and their effects on the seismic cycle. The authors were able to show that fluid overpressurization in the Alpine Fault core controlled the mechanical behaviour of the fault and could be responsible for future rupture initiation. The article is very well written and organized and provide very important data and interpretation to understand more in detail the processes, which control the seismic cycle of active fault zones. I only have minor comments on the manuscript.

General comments:

1. Portion/fraction of weak minerals related to fluid overpressure in relation to pre- existent weak minerals (clay minerals in gouge zone / fault zone) - Abstract (line 25-29) and chapter 5.3 line 245-261:

You analyses gouge material, especialy clay minerals. In chapter 2. you mentioned that the gouge material is a reworked product probably as a result of ultracomminution due to multiple shear events under brittle conditions. The local presence of authigenic smectite clays (Schleicher et al., 2015) and calcite and/or chlorite mineralization within sealed fractures and in the gouge matrix (Williams et al, 2017) indicate that mineral reactions are restricted to an alteration zone within the fault core.

You conclude that due to fluid overpressure a weak mineral phase was introduced into the fault zone.  My Question:  What portion/fraction of the gouge material is related  to the fluid overpressure and what part related to former events, e.g. ultracomminu- tion together with fluid mobility/sealing, without fluid overpressure. Because when you already deal with a weak clay-rich rock and afterwards another weak phase in fluid- overpressurized pores is precipitated (e.g. clay, graphite), than the influence of this minor third weak phase (volume weighted with respect to the existing surrounding weak fault rock) on the already existent rheology is relatively low. Please clarify this point.

Response: Thank you for this comment. Before answering your question we would like to clarify two points: (i) the conclusion that mineral phases were introduced was documented by previous studies on these rocks (i.e. Schleicher et al., 2015 and Williams et al., 2017), our data only confirm the conclusions made in those studies; and (ii) in lines 25-29 and 245-261 (now in lines 248-264), we don't state that fluid overpressure caused the precipitation of weak mineral phases. Instead, we suggest that fluid-filled pores are favourable environment for mineral precipitation, which could further reduce the already extremely low total porosities in these rocks, and thus lead to fluid overpressure. In this way, the addition of only small amount of new material phases could have a dramatic effect on the mechanical behaviour of the fault. Furthermore, if the newly precipitated material is with low frictional properties, the likelihood of a fault slip would be even higher.

2. Analytical detection of 1 *µm*-sized pores line 185-187: On figure 8b pores have sizes comparable to the small range of pores segmented on XCT images (> 1.3 *µm* in diameter), and thus we conclude that both nano- and micro pores within the Alpine Fault core are distributed on grain and phase boundaries, especially of clay minerals (Fig. 8).

MB-comment: In Fig. 8b the pores are extremely flattened /elongated and only the long g axis show a value > 1 *µm*. I am not sure if you can measure with synchrotron and a voxel size resolution of 1.3 *µm* such elongates pores where the calculated diameter is <1.3 *µm*. Please clarify this point.

Response: We do not imply we have measured the exact same pores shown on the TEM images. We only state we observe pores with comparable (but not the same) size, which allows us to suggest that the distribution of pores in these rocks is focused along grain boundaries. In response to the review by James Gilgannon, we uploaded a screenshot from Avizo that demonstrates similar distribution of pores in the micro-scale as well.

Following points are minor comments:

Line 136: High resolution TEM images

MB-comment: Actually, your TEM images are not High Resolution images. The defini- tion of high resolution TEM imaging means that you work with an Angstrom resolution in order make visible the atomic structure and that is not the case for your microstructures

Response: We will omit referring to the TEM images as high-resolution images. (Line 139)

line 200-201: To address this possibility more data for systematic analyses of pore orientations are needed

MB-comment: Please compare your observations/data with published papers, which contain similar TEM porosity analysis in clay-rich rocks.

Response: Our TEM observations are comparable with previous porosity studies in clay-rich rocks from the San Andreas Fault core (Janssen et. al., 2011) and Nojima Fault (Surma et al., 2003) zone. Both studies document grain boundary pores, that appear with irregular and/or elongated shapes. We demonstrate such pores on Fig. 8a, b and c. In the San Andreas fault rocks pores identified as inter-clay and fracture porosity were also documented and interpreted as in-situ pores (i.e. not induced by coring or mechanical damage) whenever pores are associated with newly formed clay minerals. We show pores with similar shape characteristics and mineral associations on fig. 8b and d. However, those studies focus on pore morphology and do not discusses pore orientations.

line 229-231: Thus, the comparatively lower porosity estimates of the Alpine Fault core than other active faults (e.g. the Nojima Fault, Surma et al., 2003, and the San Andreas Fault, Blackburn et al., 2009) can be attributed to the fact that the Alpine Fault is late in its seismic cycle (Cochran et al., 2017).

MB-comment: Do you refer to the latest seismic event in the year 1717 and the average seismic cycle of 291±23 years? please clarify this - because maybe the reader already forgot that you have mentioned this point at the begin of the paper.

Response: Yes, we did refer to the last seismic event in 1717, and this information was provided in lines 55, 56. However, we can repeat this here to make the manuscript easier to follow. (line 234)

Figure 1 Line 403 (Figure 1):

MB-comment: Please provide GPS data of the drilling site Figure 8 (a)

Response: GPS coordinates will be added to the caption of figure 1. (line 408)

MB-comment: I presume the dark structures are the pores - please point directly with arrow-tip onto the structures. Otherwise it is a little bit confusing, especially for readers who are not familiar in reading TEM images

Response: Figures 8a and 8c are bright-field images, where porosity appears as bright contrast areas. Figures 8b and 8d are high- angle annular dark field images, where pores appear as dark contrasts areas. The arrows are positioned on the figure accordingly. However, we will add this additional information to the caption. (lines 433-434)

Figure 8 (c)
MB-comment: where are the quartz/feldspar grains with the strain shadow? Is it the grain in the middle of the image? Than please shift the text "Ellipsoidal pores" to an area in the image where it does not cover an essential part of the image.

Response: We will modify the figure.

[revised manuscript text omitted]

---

## Referee Report (RR1)

Dear Editor,

Please find my general and specific comments below for my review of the revised manuscript by Kirilova et al., titled "Micro- and nano-porosity of the active Alpine Fault zone, New Zealand".

**General comments**

In their revised manuscript the authors have chosen to retain the manuscript largely as is. The few additions that have been made in the methods section now help the reader not make the mistake I made when first reading the initial submission.

I understand why the authors have been reluctant to change the manuscript as it is well written and has a flow that guides the reader but I do not find the answers to my last comments satisfactory in addressing the fundamental limitations of the data. The chief concern I have is that the porosity segmentation is too simplistic to allow the interpretation that follows in the discussion of a porosity and permeability gradient. In the following specific comments I have tried to clarify why I think that the authors may be over-interpreting their data and as such why I think that the authors should make room for more discussion about the limitations of the data.

To be very clear I think that the work is good and the results can be published but first there is a need for more transparency in the methodological workflow used and how it may affect any interpretations. If the authors wish to make the claims they currently make then I think these must be made more cautiously and with enough information given to allow the reader to evaluate the discussion points. As the manuscript currently stands, the results presented do not allow a gradient in porosity and permeability to be interpreted nor the further interpretation of this that variations in dissolution-precipitation must exist, which are both key discussion points of the current manuscript.

Best,

James Gilgannon

**Specific comments**

I have kept to the sections previously defined and have written my comments to the authors responses under headers of the same names. My original comments are in the lightest grey, the author's responses are in italics and an intermediate grey colour, while my reply to the author's response is in black coloured font.

**Lines 123 - 130: Pore shape descriptors**
The manuscript would significantly benefit from a figure illustrating the relevant aspects of the use of the covariance matrix. For example, I do not understand the author's characterisation of sphericity. I may have misunderstood the description but the ratio of two eigenvalues, which are both contained within a plane, surely cannot describe the deviation/tendency to a sphere, or have I misunderstood the metric you present? I am more familiar with sphericity being the ratio of the equivalent surface area of a sphere with the same volume as the pore volume over the actual surface area of the pore volume (e.g. Wadell, H. (1932))?
For this reason, I think that the section would benefit enormously from an example figure that corresponds to, and visualises the explanation of the metric. I imagine this would be best done with some specific examples of pore volumes from your data set. If the authors have not come up with the method themselves then I think that a citation for the more curious reader is also necessary.
Wadell, H., 1932. Volume, shape, and roundness of rock particles. The Journal of Geology, 40(5), pp.443-451.

*Response: Thank you for this comment. No, we have not come up with the methods ourselves. All the shape analyses we performed are based on functions embedded in Avizo software that yield volumetric and shape characteristics for each segmented material in numerical format (lines 118). We simply plotted the results on bivariate histograms by using Matlab as stated in line 125.*
*We do not find it necessary to include in the manuscript a description of how the software produces those results as Avizo software is trusted source and every user/reader can refer to their library. However, here we provide a brief explanation of the functions we have used:*
*The covariance matrix is built on the basis of the moments of inertia and can be written as: …*

*(matrix provided and implemented by Avizo software)*

*By using this matrix, the software computes the three eigenvalues by using a Singular Value decomposition. In an elongated ellipsoid the largest eigenvalue will describe the longest axis of the 3d object.*
*In this context, the deviation of the spherical form (i.e. anisotropy - a value extracted from Avizo software) is measured as 1 minus the ration of the smallest to largest eigenvalue. In a 3D object if the smallest and the longest axis are equal, the medium will have the same value as well, describing a spherical object and having numerical value = 0.*

**Reply to Response:**
I agree that the authors should not repeat the Avizo user manual in their manuscript but I was not suggesting that this was necessary. The authors already do a great job of explaining in writing the ratios they implement and as such if the authors do not wish to provide a figure for explanation then this is a choice I understand.

However, I do not think that the term sphericity should be used as this is already a term expansively used in the literature. As the authors wish to defer to the Avizo user manual then the metric should be changed to the term that it is defined as in Avizo, which the authors refer to above as anisotropy. I say this because the Avizo user manual (at least in the copy for v. 9) has a separate section on custom functions where the user is shown how to implement the well known sphericity calculation I cited in my last review. Calling a metric by a name it is not known as, and which happens to be the name of an already established metric, is confusing.

**2. Questions/concern regarding total porosity calculations**

My questions/concern is regarding the uncertainty associated with the filtering of pore data used and how this translates into the discussed differences in the magnitudes of the total porosity from different samples. Your TEM results show that very small fractures exist, which you identify as fracture porosity and, by the general argument of the paper, could have resulted from coring. While these fractures shown in fig. 8d are below the XCT resolution, I am brought to wonder how many slightly larger pores exist that are actually induced fractures. For example, the fact that so many small pores identified by XCT are almost completely flat in shape (fig. 6) might reflect that many small fractures, that are not syn-kinematic, are retained in the analysis. Therefore for me a question that presents itself is; does a simple size threshold, as you have used, have an appropriate amount of filtering information to allow a discussion about a difference of 0.14% porosity? Stated more plainly, how do you know if the variation between samples is not just a function of the degree to which each lithology experienced the coring and retrieval? Alternatively, can you rule out that the variation of 0.14% may just be related to the uncertainty of the polynomial fitting used to cap the pore size for integration?

I am uncertain if it is correct to straightway interpret this difference of 0.14% as meaningful. I think that more interrogation of this result needs to appear in the discussion before it is taken forward as independent confirmation of other literature. It might be that the authors wish to use the bore hole and laboratory measurements of permeability that are mentioned in the text to quantitatively check if the difference of 0.14% in total porosity can account for these differences in permeability.

I am aware that this would require some assumptions when calculating but it would provide a base to the interpretation that a difference of 0.14% porosity between samples is meaningful. As currently presented I think that the result only convincingly shows that each calculated porosity is of the same order of magnitude.

*Response: We understand the reviewer's concern that big pores and small fractures could get easily misinterpreted/mislabeled in XCT datasets. This exactly is the prime reason why we decided against calculating total porosities in these samples by simply using 'connected components' and instead we fitted the data to a polynomial curve (mentioned in lines 115-123). We believe that implementing a mathematical approach is much more trustworthy than limiting the data based on the interpreter's bias. Furthermore, our total porosity calculations (by using the polynomial fit) roughly coincide with the total porosities yield by calculating the total porosities based on connected components with up to 200 voxels. You can see these numbers on the table below:*

*The reviewer also expressed concern about the fact that some of the very flat pores may represent fractures. We acknowledge the validity of this statement. However, we believe that our approach of excluding cracks is efficient and possibly the best methodology for analyzing these samples (i.e. fitting the data to a polynomial curve). Furthermore, the shape of these pores is also very likely to result from their distribution along grain boundaries, especially of clay minerals (lines 190). The authors of the manuscript are in favour of this second scenario.*

*And last but not least, the difference of 0.14% of total porosity in these samples may seem insignificant to the reader. However, all of the samples contain extremely low porosities, and thus only 0.14% more pores actually result in doubling the amount of pores in sample DFDP-1B 69_2.57 in comparison to the rest of the samples. Thus, a discussion here is not only meaningful but also required, and very well related to changes in lithology in between the samples (lines 219-226, now in lines 222-229), and previous permeability measurements of these rocks (lines 233-236, now in lines 236-240).*

**Reply to Response:**

Thank you for your answer and considering my concerns. From your reply I take away that you feel the use of the polynomial fit leads to a more trustworthy total porosity calculation. I am not sure that I agree that there is much difference between using the 'connected components' method vs the polynomial fit of pore data.

The first reason I do not agree is that:

1. They are both just pore 'size' filters. The connected components caps the total porosity to the pores below 200 voxels and the integration of the polynomial simply caps the total porosity to pores below those deemed as the largest by the x intercept.

The fact both methods produce similar total porosities (cf. table in your reply to my comments) probably just reflects that the size filtering is similar and there are not so many pores between the 200 voxels cap (~430 $\mu m^3$) and the higher polynomial caps (~580,~680, ~860, ~760 $\mu m^3$).

The second reason, which I think is more problematic, is:

2. A simple grey scale threshold has enormous uncertainty associated with it and this uncertainty can account for variations on the order of those that are reported (i.e. doubling/halving).

This choice of segmentation method would affect the outcome of any later data filtering, be that using the connected components calculation or the polynomial fit.

For the purposes of illustration I have constructed a fake porosity microstructure (fig. 1) to quantitatively highlight that the choice of threshold can give a range of values, in some cases the variation can be as much as double. This fake porosity microstructure has been constructed to have ~2% porosity (fig. 2a). When one chooses three different thresholds to segment the pores by (fig. 2b and c) one gets three different values in total porosity (fig. 3). One can see in figure 2c that all of the chosen thresholds are reasonable for the fake pores and their resolution, but the difference between those choices is as much as double the total porosity (1.1% vs 2.1%).

In your methods you have not reported the thresholds chosen or how data between samples looks (for example, are different core's XCT data of similar quality or not?, and how do their microstructures compare in XCT?, how do the greyscale histograms, and hence thresholds, compare between samples?). From the 2D sectioned images you provided in your reply to my comments I can see that the small pores are made of not very many pixels in a slice, which makes them susceptible to the sensitivity I have highlighted in my quantification of the fake microstructure. Therefore it is very hard to believe in the significance of a 0.14% difference in your samples.

In my original review I suggested that if you wanted to pursue the claim that the 0.14% difference is significant you would need to either bring compelling microstructural evidence to support such a claim or qualify if the porosity variation you report can account for the permeability gradient reported by other studies that you cite. As neither of these, nor any other supporting line of argument, has been included I think that you cannot consider this difference beyond the uncertainty of the method used. The consequence of this is that as the results stand it is erroneous to discuss the observed differences as independent verification of a permeability gradient reported by other studies or to infer that there is more or less dissolution-precipitation processes active in one sample over the other.

[Figure]

Figure 1: Generating a fake porosity microstructure. (a) ~200 pixels that represent pores were randomly generated in a 100x100 matrix. This represents a total porosity ($\phi$) of ~0.02. The simple single pixel pores were then resampled to increase their resolution (b) and then a greyscale gradient along pore edges was introduced by applying a Gaussian blur to the image (c).

[Figure]

Figure 2: Segmenting the fake porosity microstructure with three different threshold values. (a) is the same microstructure as shown in fig. 1c. The histogram of pixel values in fig. 2a is presented in (b). Alongside this three different thresholds to segment porosity are shown in fig 2b. These thresholds are visualised for example pores from fig. 2a in (c).

[Figure]

Figure 3: The total porosity calculated with three different threshold values.

---

## Author Response (AR2)

Dear Dr Kirilova and coauthors,

I have now carefully read your revised paper and also considered James Gilgannon's review of that version of your manuscript and came to the conclusion that, in its current form, the manuscript is not yet ready to be published in SE.

I do think though that with a rework that addresses the concerns, your paper will make a robust and well-cited contribution, and I look forward to receiving your revised version.

Best regards,
Florian Fusseis

Dear Dr Fusseis,

We are very grateful for your time and constructive feedback. We acknowledge your expertise in porosity analyses and CT-data interpretation. Our interest is to produce a valuable contribution to the scientific society, thus we have carefully considered all of your comments. Please find our responses in green text below.

Kind Regards,
Martina Kirilova
On behalf of co-authors:
Virginia Toy, Katrina Sauer, François Renard, Klaus Gessner, Richard Wirth, and Xianghui Xiao

\*\*\*\*\*\*\*\*\*\*\*\*\*\*\*\*\*\*\*\*\*\*\*\*\*\*\*\*\*\*\*\*\*\*\*\*\*\*\*\*\*\*\*\*\*\*\*\*\*\*\*\*\*\*\*\*\*\*

I see a certain disconnect between the data you present and your discussion, in particular Section 5.3, in particular, it isn't obvious to me that your data really support the conclusions you draw there. This doesn't mean that the conclusions are incorrect, but rather that the trends in the data you report are not overly obvious.

In particular, I have concerns about the validity of the shape analysis, which hinges entirely on the covariance matrix. The matrix really describes the shape of an ellipsoid, not a pore. I would argue that for the conclusions to be robust, one would need to demonstrate that the eigenvalues/vectors indeed are a good representation of pore shapes. Pores in deformed rocks can have very convoluted shapes which are difficult to describe with standard mathematical operations. There are, however, more sophisticated tools, such as Minkowski funtionals, which you could use.

Response:
We realize that it is possible to characterize shapes of non-ellipsoidal features using more complex methods than fitted ellipsoids. However, it is not true that the covariance matrix approximates the pores as ellipses. Instead, the covariance matrix can be calculated on any shape. For an ellipse, the 3 eigenvalues will represent exactly the 3 axis of the ellipse. For more complicated shapes, the 3 eigenvalues measure length scales of the object in 3 perpendicular directions of space. In addition, pores detected in our samples are predominantly ellipsoidal (see figure below).

Furthermore, fitting ellipsoids is a standard and widely employed method in the field of porosity analysis. For example, Menegon et al., (2015) used fitted ellipsoids to estimate the length of the longest axis of pores in monzonite ultaramylonites. The editor is a co-author of this manuscript. This means our data are easily compared to measurements made in prior studies. In addition, Sufian et al., (2018) noted that their pore orientation tensor (which is the eigenvectors of a fitted ellipsoid) is of a similar form to the Minkowski interface tensor. Thus, we do not think that undertaking the analysis with Minkowski functionals as recommended will substantially change our results. We have not performed a new analysis.

[Figure]
 Ellipsoidal pores in sample DFDP-1B 69_2.54 (Fig 8a in the manuscript)

My second major concern arises from the polynomial fitting - where really the R^2 values aren't too great. I suspect, on the basis of my own experience, that you could yield substantially different porosity values quite easily if you used slightly different fits.

Response:
We acknowledge the concern that R^2 values are "not too good". The method we used is designed to estimate and exclude fractures within the sample data, which means the R^2 will be low. We can observe microfractures present even at small volumes and this method allows us to estimate when those fractures begin to adversely affect a total porosity calculation. We are not optimizing for R^2, we are biasing the fit to smaller volumes where more pores exist, and biasing the fit against higher pore volumes where more fractures exist. A consequence of this is that the R^2 values are lower, but we believe it results in a more accurate estimate of total porosity excluding fractures.

Specific comments (line numbers)

I think it might be helpful if you very briefly, as obviously described in detail elsewhere, mentioned how these samples are related to the activity of the fault. A figure (BSE images and corresponding slices through uCT data) showing the microfabrics of the four samples should be added (and can replace some of Figs 1 and 2). Also, mineral compositions, grain sizes and proportions should be reported in a table. This is relevant for your section 5.2 below.

Response:
Section 5.2 discusses our data in the context of published studies where all the information required by the editor is described in detail. However, images of the samples are available as part of an unpublished PhD thesis (it is currently in final revision stage). We could include them but would then want to add the student's name to our author list. Is that an acceptable solution?

Please report on the sample-detector distance, as this is important to estimate a possible phase contrast in the data.

Response:
The sample detector distance was 70 mm. Information added in line 105.

The sample size is only of minor relevance here, since the field of view will only be just over 2 mm high at 10x magnification (which I believe gives 1.3 um voxel size). What is the widths of your FoV?

Response:
The width of FOV was 2.81 mm. Information added in line 105.

Why no phase retrieval in reconstruction?

Response:
These samples were analyzed in 2011 before such methods were implemented at APS.

What are the dimensions of the sub-samples, how did you choose them, and are they statistically relevant?

Response:
The subsamples were cropped from representative areas of the whole sample that show typical mineral associations and porosity distribution within a sample. The size of the sub-samples ranges between $601^3$ voxels and $801^3$ voxels. Differences in the sub-sample sizes depend on the quality of the whole sample (e.g some samples were cut smaller to exclude artifacts or big fractures, which were obviously induced by coring and/or sample preparation).

Why use a primitive thresholding tool, where much more sophisticated tools for local crack detection are available (e.g. Voorn et al., 2013, 2015, Ma et al., 2020)? This is especially relevant since the entire paper hinges on the analysis of your segmented porosity.
Voorn, M., Exner, U., & Rath, A. (2013). Multiscale Hessian fracture filtering for the enhancement and segmentation of narrow fractures in 3D image data. Computers & geosciences, 57, 44-53.
Voorn, M., Exner, U., Barnhoorn, A., Baud, P., & Reuschlé, T. (2015). Porosity, permeability and 3D fracture network characterisation of dolomite reservoir rock samples. Journal of Petroleum Science and Engineering, 127, 270-285.
Ma, X., Kittikunakorn, N., Sorman, B., Xi, H., Chen, A., Marsh, M., ... & Skomski, D. (2020). Application of Deep Learning Convolutional Neural Networks for Internal Tablet Defect Detection: High Accuracy, Throughput, and Adaptability. Journal of Pharmaceutical Sciences.

Response:
We are very aware of the persisting problems with crack/fracture analyses by using simple thresholding tools. However, this study does not focus on crack/fracture detection, but instead aims to exclude any cracks/fractures from the porosity analyses. In addition, pores in our samples appear as high contrast very dark grey-scale materials. In this case, the binary thresholding tool implemented by Avizo software is more than sufficient for robust porosity segmentation. Similar approaches have been previously used by multiple studies, including Menegon et al., 2015 and Gilgannon et al., 2017, where the editor is a co-author.

You also must have labelled the data, which degree of connectivity did you choose? This will significantly influence your pore statistics.

Response:
We assume the editor is concerned about the connectivity typically manipulated within the morphological operation "connected components" (i.e. "face" vs "edge" connectivity between voxels). We used this operation purely for visualization purposes (lines 120- 122). This limitation of the data was not implemented at all in the methodology for total porosity estimates. Thus, it does not affect our results. Nonetheless, this information is added in line 116.
"To omit the cracks, thresholded components with volumes larger than the volume of 200 face connected voxels (439.4 µm3) were excluded from the binary label images"

119-128 This approach could be discussed considering the findings of Keller et al., 2011, 2013a (and others) on the porosity distribution in clays (as these being related to gouges).

Keller, L. M., Holzer, L., Wepf, R., & Gasser, P. (2011). 3D geometry and topology of pore pathways in Opalinus clay: Implications for mass transport. Applied Clay Science, 52(1-2), 85-95.

Keller, L. M., Holzer, L., Schuetz, P., & Gasser, P. (2013). Pore space relevant for gas permeability in Opalinus clay: Statistical analysis of homogeneity, percolation, and representative volume element. Journal of Geophysical Research: Solid Earth, 118(6), 2799-2812.

Response:
Thank you for the relevant references. Both studies are valuable contribution to the field of porosity analyses. However, they use only SEM and FIB data and do not integrate CT-data in their studies. Consequently, their approach is very different from ours and also their analyses mostly focus on pore size distribution and pore geometry. Instead, here we focus our interpretation and conclusions mainly on total porosity estimates. In our manuscript, we propose a method for total porosity calculations that in mathematically robust approach estimates total pore volumes while excluding cracks and fractures from the overall percentage. Excluding cracks from our datasets is fundamental because we are dealing with drill-hole samples that have suffered fracturing during exhumation.

"with limit of" is misleading, is this smaller or larger than 200 voxels?

Response:
Thank you. The text was modified to "limited up to 200 connected components"

10 voxels as a lower limit is too small, consider the limited number of shapes you could build out of 10 cubic lego bricks - this inevitably introduces a bias in your analyses. Consider a lower limit of 5^3 voxels.

Response:

The number of shapes you can build out of 10 cubic bricks according to the sequence which defines the number of 3-dimensional polyominoes (or polycubes) with n cells  is 346543, which is sufficient granularity for determining shape and orientation without bias. (N. J. A. Sloane, A Handbook of Integer Sequences, Academic Press, 1973)

In addition, introducing a lower limit of 5^3 voxels in our datasets (voxel size of 1.3 microns^3) will result in materials larger than 274.625 microns^3. Therefore, vast amount of materials of interest will be excluded. Furthermore, the shape and orientation analyses may be biased by the prevailing influence of cracks with this size.

Regardless, here we present comparison between lower limits of 10 voxels and 5^3 voxels. We use as an example sample DFDP-1B 69_2.54. The figures below show that a lower limit of 5^3 (i) does not influence the pore orientation patterns; (ii) excludes  vast majority of our segmented materials which are of interest, and thus introduces bias in the resulting elongation vs pore size histogram.

[Figure]

Stereonet with lower limit of 10 voxels (from figure 7)

[Figure]

Stereonet with lower limit of 5^3 voxels

[Figure]

Elongation vs pore volumes (from fig. 4) with lower limit of 10 voxels

[Figure]

Elongation vs pore volumes with lower limit of 5^3

135-137 I disagree with the statement that the eigenvalues describe the shape of a pore - they describe the shape of an ellipsoid, which only under fortunate circumstances will be a good approximation of the actual pore shape. To make conclusive statements on the shapes of your pores, you need to go a step further and use, e.g., Minkowski functionals (Arns et al., 2002, Lehmann et al., 2006, Wildenschild & Sheppard, 2013).
Arns, C.H., M.A. Knackstedt, and K.R. Mecke. 2002. Characterizing the morphology of disordered media. p. 37–74. K. Mecke and D.
Stoyan (ed.) In Morphology of condensed matter: Physics and geometry of spatially complex systems. Lecture Notes in Physics.
Springer, Germany.
Lehmann P et al. Tomographical imaging and mathematical description of porous media used for the prediction of fluid distribution. Vadose Zone J 2006;5(1):80.
Wildenschild, D., & Sheppard, A. P. (2013). X-ray imaging and analysis techniques for quantifying pore-scale structure and processes in subsurface porous medium systems. Advances in Water Resources, 51, 217-246.

Response:
The recommended papers deal with datasets where the porosity is connected. In these studies, shape analyses using covariance matrix decomposition would not work, thus they needed to implement other techniques. However, pores in our data are separated and the covariance matrix is calculated on each single pore. To clarify this in line 133 the text was modified to:
"Individual pores in our dataset are separated (Fig. 2c), thus the covariance matrix of each pore was calculated, and the three eigenvalues of this covariance matrix were extracted."
Also, as we previously responded to the editor's major comment, the covariance matrix can be calculated on any shape. Thus, our analyses are relevant, and the use of different methods would not benefit the manuscript.

155-157 The screenshot you have added in your rebuttal to J Gilgannon indicates that your segmentation captured only a very small proportion of the actual porosity. How have you chosen the threshold value for your segmentation?

Response:
We acknowledge that the image provided has a low resolution (due to Avizo software screenshot resolution) and may not show the threshold range properly. But we strongly disagree that we have not captured all the porosity in these samples. Pores were identified as the darkest grey-scale materials on images with good contrast, consequently thresholding was not challenging. We performed thresholding by selecting the corresponding grey-scale range manually and individually for each sample due to slight brightness contrast differences in between samples. To ensure all pores were captured the selected threshold range was inspected throughout the 3D stack before thresholding being applied.

164-167 Fig 7 - what is the reference frame for these stereoplots? I see you are speculating that they trace a foliation, but obviously the maxima are all roughly at the same orientation between the plots.

Response:
In these lines we do not talk about foliation. Here, we only describe the observed maximas:
"The orientations of the individual pore units show two distinctive peaks with opposite vergence, defining bipolar distributions of pore orientations"
Later in the manuscript, we only speculate that pore orientations reflect the orientation of clay minerals. The assumption is based on the predominant distribution of pores along grain boundaries of mainly clay minerals, and the abundance of intercaly pores (lines 200-206). The similar orientation of the observed orientation maxima in all samples is due to the fact that samples were drilled parallel to the foliation. This information is added in line 104.

- "the gouges have composition..." - rephrase

Response: changed to "are composed of"

- What are the criteria by which you distinguish cracks from other types of pores in your TEM data? It is not really evident from Fig. 8.

Response:
Fracture porosity was distinguished in quartz-feldspar domains (fig. 8d) where elongated, thin void spaces are associated with multiple grains and occasionally disrupt grain boundaries. The information was added to lines 178-179.

[Figure]

185-187 This statement is incorrect, since this is the voxel size, which does not equate to the spatial resolution.

Response: Thank you. We now say "acquisition constrains" instead of "resolution constrains"

188-190 I think this needs to be contrasted with Keller et al (2013b)'s study on Opalinus Clay. Obviously, they investigated undeformed clay as opposed to highly sheared fault gouge, but the key finding suggests that, in clay-mineral rich portions, there may be a very substantial volume of porosity hidden. What are your arguments in support of this statement?
Keller, L. M., Schuetz, P., Erni, R., Rossell, M. D., Lucas, F., Gasser, P., & Holzer, L. (2013). Characterization of multi-scale microstructural features in Opalinus Clay. Microporous and mesoporous materials, 170, 83-94.

Response:
Thank you for this reference. We acknowledge the assumption that nano-scale porosity would not significantly influence the total porosity estimates is not robust. Thus, this statement was omitted from the manuscript.

(also section 3.4) How were the locations for the FIB foils chosen? How did you ensure they were representative?

Response:
The Alpine fault samples, derived during DFDP-1, have been subjected to intense microstructural analyses both onsite and post drilling operations by the DFDP-Science team. Consequently, numerous studies have described these rocks in detail (e.g. Toy et. al., 2015; Schleicher et al., 2015 ). Based on these previous studies and our own observations we believe we have selected areas of interest for FIB foils that yield representative results.

192-194 I don't understand how the first half of the sentence relates to the second, and how you arrive at that conclusion.

Response:
For better clarity the text has been modified, and now says:
"The pores visible on grain and phase boundaries in Figure 8b have similar sizes to the  pores segmented on XCT images (> 1.3 µm in diameter), thus we conclude that this is the typical habit of both nano- and micro-pores within the Alpine Fault core (Fig. 8)."

205-208 cf. Keller et al., 2013b

Response:
We do not think that this is an appropriate reference to cite here. No change made.

Please restructure this section for clarity and revise paragraph structure. The reader shouldn't have to read until the end to figure out what this is about.

Response:
We have restructured the section accordingly.

213-214 How do you identify these clay minerals as authigenic?

Response:
We interpret these clays as authigenic because of the coexistence of newly precipitated fine materials and coarser grains, previously documented in detail by Schleicher et al., 2015.

214-216 Annotate these observations on the figure, and pls add references to support the pressure solution argument.

Response:
The figure is now annotated. Reference added.

Would be good if you provided grain size distributions for the two materials, see comment for line 98.

Response:
Grain size distribution is available as part of an unpublished PhD thesis (it is currently in final revision stage). We could include them but would then need to add the student's name to our author list. But note that substantially longer method documentation will be needed to report them here.

Does your analysis really demonstrate significantly different porosity values between the two cataclasite units, given the R-squared values of your regression model?

Response:
As we previously responded to the editor's second major comment, the method used here is intentionally attempting to estimate and exclude fractures within the sample data by biasing the fit which yields lower R^2. Thus, the results presented here are robust and do demonstrated significantly different porosities. In addition, as we mention in response to reviewer James Gilgannon, similar porosity differences were also observed by calculating porosities after implementing the morphological operation "connected components".

240-251 This paragraph is a bit bold on the assumptions. First, low porosity is not necessarily consistent with low permeability, and I don't see how you verify permeability measurements? Aren't you overinterpreting a difference in porosity of 0.12%, given the uncertainties outlined in your own statistics plus the uncertainties introduced by the data processing steps and sub-sampling sites you have chosen?

Response:
We agree that porosity is not necessarily consistent with permeability and acknowledge our bad choice of wording in this paragraph. Here, we only aim to compare our porosity data with published permeability measurements on these rocks, which show similar trend. We also indicate that our porosity estimates are in agreement with previously documented permeability gradient in these rocks. The text was modified, and we now say:
"Our data thus provide independent verification is comparable with of the permeability measurements in that study (Carpenter et al., 2014) and yields increased confidence in their interpretation of a permeability gradient with distance from the PSZ"
We strongly disagree we are overinterpreting our results, because as previously described our R2 values do not introduce uncertainties in our data (see the comment above).

If it is precipitation of hydrous minerals in pores (as opposed to veins), how much does the pore fluid pressure actually change as clay minerals form?

Response:
True, some of the fluids in the pores will be included in the newly formed clays. However, because these minerals also have other components (e.g. silica), the total volume of fluid involved in their formation will not be sufficient to stop a general increase in pressure as pore size reduces. We have added a sentence 'However this pressure increase will be slightly offset by inclusion of fluids into new hydrous minerals' (in line 263)

Neither Byerlee nor Sibson do actually talk about precipitation in pores (but of course about fluid pressurization), could you reformulate this slightly to avoid that impression?

Response:
 We modified the text to avoid misleading references. We now say:
"(i) very small decrease of these critically low total porosities due to mineral precipitation would cause fluid pressurization, which is a well-known fault weakening mechanism described by Byerlee, 1990 and Sibson, 1990" (now in line 263).

Is there graphite present in these samples? - It isn't highlighted in Fig 8. If not, please clarify this statement.

Response:
Graphite is not shown on the TEM images presented here. However, the presence of graphite within the Alpine Fault rocks has been previously documented and described by Kirilova et al., 2017. We modified the text to clarify this. In line 272 we now say
"which was previously documented in these rocks by previous studies (Kirilova et. al., 2017)"

Fig 2 isn't particularly helpful, since it doesn't show a workflow as such (sensu flow chart). I wonder whether 2.5D "thick slices" (see, e.g. Fig. 3 in Menegon et al., 2015, Geology) would be more better.

Response:
The aim of this figure is to show that the threshold range needed to segment all pores also captures cracks, and thus the later need to be removed for total porosity estimates and visualization purposes. Thus, the figure achieves the purposes it was designed for.

Fig caption "after the fractures" - "after removal of the fractures from the segmented data".

Response: The text was modified.

Fig 2c, since this is a perspective figure, the scale bar does not apply to the entirety of the image, pls consider/correct.

Response:
Thank you for this comment. We acknowledge the mistake on our side thus we removed the scale.

438/439 I don't see a difference between pores in b) and d). What are your arguments for the pores in d) being fractures?

Response:
We acknowledge that the morphology of pores in (b) and (d) may appear to the reader as fairly comparable. However, (b) represents a phylosillicate-rich gouge area, where very elongated, extremely thin pores occur parallel to phyllosilicates orientation, thus we interpret those as "interclay porosity". Whereas in (d) pores are in association with multiple quatz-feldspar grains and occasionally affect grain boundaries. Both interclay and fracture pores with similar characteristics were observed within San Andreas fault gauges (Janssen et al., 2011)

[revised manuscript text omitted]

---

## Author Response (AR3)

Dear Dr. Fusseis,

Our paper makes a very important conceptual contribution, in that it highlights how critically low porosity, combined with evidence that solution-precipitation occurred, means a fault is in a critical state. This may be the control on earthquake recurrence interval of many similar faults. We think that your focus (mirrored and thus doubly weighted by the reviewer, who we note was in the past a member of your research group) on procedural details of the analyses obscures the major contribution of our paper. As demonstrated below, even if we use a different analytical method, the key data that stimulate this concept, which is that the porosity is critically low, will not change. We hope that it is possible to allow this observation inspired concept to be presented to our community, since we expect it to stimulate new and novel research in other fault zones.

Following your request, we have included SEM data to the manuscript. This data is a part of an unpublished PhD thesis, thus we have added that thesis' author, Dr. Risa Matsumura, as a co-author of this manuscript.

Below we have provided a point-by-point response. Line numbers are listed with respect to the manuscript with track changes on.

Best Regards,

Dr. Martina Kirilova

On behalf of co-authors: Virginia Toy, Katrina Sauer, François Renard, Klaus Gessner, Richard Wirth, Xianghui Xiao, and Risa Matsumura

Dear Authors,

Thank you for your reply to my comments. Unfortunately, it does not convince me that your porosity estimates are robust and support your interpretations sufficiently. Your descriptions are neither precise nor detailed enough to allow for a reproduction of your analysis, and I do miss evidence that supports the choices you have made in your data processing. Just a few examples to justify these statements:

What data support your claim that binary thresholding is "more than sufficient for robust porosity segmentation"?

Response: See our answer to point 1 in Dr. Gilgannon's review.

What threshold values did you use, and how were they chosen? Where they the same between all four datasets?

Response: See our answer to point 2 in Dr. Gilgannon's review.

What is the effect of the NLM filter on your porosity quantification?

Response: Non-local means filter reduces noise, and several studies have employed it (e.g. Thomson et al., 2018; Renard et al., 2019). We have now added two sentences at the beginning of section 3.3 that summarize the effect of this filter.

Are your subvolumes statistically representative?

Response: The sub-volume crops in each sample are representative. They were carefully chosen to (i) preserve the typical microstructural characteristics within a sample, and (ii) to exclude ring artifacts or big fractures (obviously induced by coring) as much as possible.

Where all of your pores labelled on the basis of face-connected pores?

Response: All pores were labelled on the basis of face-connected pores. This question was already addressed during the previous round of revisions and the information was added at line 116, but due to further revisions it is now at lines 143 and 147.

What degree of voxel connectivity did you choose to label your pores?

Response: No limit of voxel connectivity was implemented. Limits within Avizo software were introduced purely for visualization purposes as previously noted at line 147.

All of this is critical, as your entire interpretation is based on your segmented porosity data. Your reference to Menegon et al., (2015) and Gilgannon et al. (2017) is to a degree justified, both do indeed use the same thresholding technique. However, we now have significantly more sophisticated algorithms freely available, and they should be used. The significance of the choice of segmentation technique has been demonstrated in a large number of publications.

Response: See our answer to point 1 in Dr. Gilgannon's review.

On the basis of these, I am actually certain that I could analyze your data with a different set of algorithms and arrive at significantly different numbers, which, I think, highlights the principal problem.

Response: See our answer to point 1 in Dr. Gilgannon's review.

We would be interested in the opinion of the Editorial Board whether this statement is in line with the code of conduct of Solid Earth (point 3 in https://www.solid-earth.net/policies/obligations_for_editors.html: "*An editor must respect the intellectual independence of authors*").

As is, I cannot recommend the paper for publication. I do you encourage you to address the concerns raised above, and I would further ask you to consider the review of your revised manuscript provided by James Gilgannon.

Response: Because the second review of Dr. Gilgannon was not submitted as part of the interactive discussion process, all the authors failed to recognize the submission of this review. We now provide a response to Dr. Gilgannon's comments.

I would further ask you to include the detailed microstructural sample description that must form the basis for your interpretation of the porosity in these rocks.

Response: We have added section 3.4, lines 214-220 in section 4.2 and Figure 8.

-------------//-------------//-----------//---------//------------//------------//------------//------------//----

Dear Editor,

Please find my general and specific comments below for my review of the revised manuscript by Kirilova et al., titled "Micro- and nano-porosity of the active Alpine Fault zone, New Zealand".

General comments

In their revised manuscript the authors have chosen to retain the manuscript largely as is. The few additions that have been made in the methods section now help the reader not make the mistake I made when first reading the initial submission.

I understand why the authors have been reluctant to change the manuscript as it is well written and has a flow that guides the reader but I do not find the answers to my last comments satisfactory in addressing the fundamental limitations of the data. The chief concern I have is that the porosity segmentation is too simplistic to allow the interpretation that follows in the discussion of a porosity and permeability gradient. In the following specific comments I have tried to clarify why I think that the authors may be over-interpreting their data and as such why I think that the authors should make room for more discussion about the limitations of the data.

To be very clear I think that the work is good and the results can be published but first there is a need for more transparency in the methodological workflow used and how it may affect any interpretations. If the authors wish to make the claims they currently make then I think these must be made more cautiously and with enough information given to allow the reader to evaluate the discussion points. As the manuscript currently stands, the results presented do not allow a gradient in porosity and permeability to be interpreted nor the further interpretation of this that variations in dissolution-precipitation must exist, which are both key discussion points of the current manuscript.

Best,

James Gilgannon

Dear Dr. Gilgannon,

We are grateful for the overall positive and thorough feedback provided here and address the remaining concerns below.

1)      the porosity segmentation is too simplistic

Response: We acknowledge the efforts of the reviewer to support his views with the analysis of a synthetic case where images with known porosity are generated. These images are created with a two-step process: 1) an image of known porosity is created, 2) then this image is blurred, an effect that we consider similar to adding noise. Finally, different gray level thresholds are applied and a variability on the estimated porosity is calculated. This example illustrates very well that when the level of noise in an image is increased, a variability in the results of a segmentation technique is encountered (Andrew, 2018). In analysing our data, we proceeded in the opposite way. First, we reduced the noise in the images by applying a non-local-means filter, and then we applied a segmentation technique. This approach ensures that the differences between several segmentation techniques remain small when the level of noise in the images is small (as shown by Figure 3h of Andrew, 2018).

The main limitations of all the possible segmentation methods are data resolution, noise level, and spatial complexity of the materials of interest (e.g. Iassonov et al., 2009; Andrä et al., 2013; Bultreys et al., 2016). Furthermore, the ground-truth/real/absolute porosity is unknown in natural samples imaged with XCT (Hapca et al., 2013), unless measured independently. The thresholding method performs as well as other segmentation techniques (e.g. classification using a supervised machine learning algorithm, watershed) when the level of noise in the images is low, as demonstrated by Figure 3h of Andrew (2018). To our knowledge, there is no segmentation procedure that can be automatically applied to various kinds of samples, without the intervention of a real scientist who can adapt the segmentation procedure to the data set and even then, the estimated porosity will only be an approximation. The method we have chosen is robust and widely used in rock physics (e.g. Iassonov et al., 2009, Fusseis et al., 2014; Qi et al., 2018; Xing et al., 2018; Macente et al., 2018; 2019; Renard et al., 2019) and potential caveats on the choice of the value of the threshold have been identified in other studies (e.g. Andrä et al., 2013).

We identified porosity as the darkest phase on the analyzed synchrotron high-contrast grey-scale images. These images have less noise than data acquired in laboratory (e.g. desktop CT tomographs). In addition, porosity in these samples is represented by separated individual pores, so there were no morphological complications with respect to spatial resolution (e.g. separating fractures such as in Figure S1 in Zhao et al., 2020). Thus, it was straightforward to segment pores by binary thresholding.

More 'complex' segmentation methods could be applied as an exercise, similar to what was done by Andrew (2018). This exercise is out of the scope of the present article and would be a study in itself.

However, because we consider that the reviewer comments necessitate modification of our manuscript, we have added several paragraphs in section 3.3 to justify the choice of our segmentation technique and added two sentences in section 4.1 to estimate a variability of the porosity. We have also added some information on the variability several segmentation techniques could introduce to the estimated porosity: our preferred segmentation procedure indicates porosities in the range 0.1-0.24%. Including 20% variability due to various segmentation techniques (Andrä et al., 2013) would modify this range to 0.08-0.29%. If one considers that the level of noise in the data is low so the analysis of Andrew applies (Figure 3h, 2018), the variability between segmentation procedures would be negligible.

In summary, as we now explain in Section 3.3, different segmentation procedures will not sufficiently modify our results, and thus will not affect the discussion and central conclusions made in the manuscript.

2)      need for more transparency in the methodological workflow

Response: We strive to be as transparent as possible about our analyses. Thus, we intend to provide the analyzed datasets in a data repository (most likely via GFZ Dataservices) to accompany the manuscript when it is published. The applied gray level threshold ranges for each sample are different because of a difference in darkness contrast between the samples. The threshold ranges applied to each samples are as follows: DFDP-1B 69_2.54: 0 to 28; DFDP-1B 69_2.57: 0 to 44; DFDP-1B 58_1.9: 0 to 74; DFDP-1B 69_2.48: 0 to 46. On the screenshots below, we display the result of the selected threshold value in sample DFDP-1B 69_2.54, slice 363. This demonstrates that most of the porosity is due to fractures that formed during sample unloading, representative pores are shown with arrows.

Furthermore, to demonstrate the need for polynomial fitting in order to estimate total porosities in these samples we included further explanations in the text at lines 139-161 and an additional table in Supplementary material 1.

[Figure]

3)	the results presented do not allow a gradient in porosity and permeability to be interpreted nor the further interpretation of this that variations in dissolution-precipitation must exist, which are both key discussion points of the current manuscript

Response:

Our main conclusions are based on the fact that the total porosities in these samples are extremely low. Even if different segmentation techniques would double these porosity estimates (e.g. Andrä et al., 2013), these numbers will remain very low, well under 1% total porosity. The fact than one sample contains twice the total porosity of others is an observation that deserves discussion. We have further clarified this in the manuscript in lines 289-293. Furthermore, evidence for pressure-solution process in these rocks have been documented in numerous previous studies (Sutherland et al., 2012; Toy et al., 2015;

Schleicher et al., 2015; Williams et al., 2017) and also observed on the TEM images presented in the current manuscript. Therefore, our results are sufficient to support the main conclusion. This conclusion, which is novel and innovative, is not dependent on the absolute value of the segmented porosity.

[revised manuscript text omitted]

---

## Author Response (AR4)

Comments to the Author:
Dear Dr Kirilova,

Thanks for the submission of your revised manuscript and supplementary material, which I have read carefully.

I would like to remind you that the choice of the appropriate reviewer lies with the topical editor, and the choice of Dr Gilgannon was entirely justified, irrespective of the fact that he used to work in my group. I would also like to emphasise that Dr Gilgannon has contributed two extremely constructive reviews, both of which were available to you through the MS records website (the latter on August 6).

I understand that you feel that the review process focussed overly on technicalities, however, as described in SE's review criteria (https://www.solid-earth.net/peer_review/review_criteria.html), before a manuscript can be accepted for publication, the scientific methods presented there must be [...] clearly outlined (4), and the description of your work must be sufficiently complete to allow reproduction (5). I am satisfied that this is now the case, and will recommend your manuscript for publication pending a minor technical addition:

Given that you have chosen four substantially different threshold intervals for four different uCT datasets (DFDP-1B 69_2.54: 0 to 28; DFDP-1B 69_2.57: 0 to 44; DFDP-1B 58_1.9: 0 to 74; DFDP-1B 69_2.48: 0 to 46), I would ask you to include screenshots for all four samples with the segmented porosity highlighted (as you did for your sample DFDP-1B 69_2.54 in your rebuttal letter) in the supplementary material. I would ask you to further include the grey value histograms, showing the threshold value, with this figure.

Lastly, as I have done already in email exchanges with your coauthors Toy and Renard, I would like to apologize for the slightly irritating tone of my last communication.

With kind regards,
Florian Fusseis

Dear Dr. Fusseis,

Thank you for the time you spent reviewing our manuscript. As requested, we have added one figure with four screenshots in the supplementary material section.

Kind Regards,

Martina Kirilova

[revised manuscript text omitted]
 in association with white mica, floating within fine matrix material. (b) Rewordked cataclasite clasts in phyllosilicate-rich layermatrix. (c) CFine chlorite and white mica fillingsaggregates in between quartz clasts. (Qtz = quartz, Wm = white mica, Chl = chlorite).

[Figure]

**Figure 8Figure 9.** Transmission electron microscopy images collected  fromon the gouge sample DFDP-1B 69_2.54 (PSZ-2). (a) and (c) are bright-field (BF) images, where porosity appears as bright contrast areas. (b) and (d) are high-angle annular dark field (HAADF) images, where pores appear as dark contrasts areas. (a) TEM bright-field image of homogeneous fault gouge area. Quartz/feldspar grains, wrapped by fine authigenic clays, displaying fringe morphlogies. Pores with sub-angular shape distributed along grain boundaries. (b) HAADF image of phyllosilicate-rich gouge area. Co-existence of fine authigenic clays with coarser clay mineral grains. Elongated pores and interlayer porosity. (c) TEM bright-field image of ellipsoidal pores in phyllosilicate-rich areas. Examples of strain shadows along quartz/feldspar grains. (d) HAADF image of fracture porosity along grain boundaries of quartz/feldspar grains.